# A synthetic protein as efficient multitarget regulator against complement over-activation

Natalia Ruiz-Molina [1], Juliana Parsons [1], Madeleine Müller[1], Sebastian N. W. Hoernstein [1], Lennard L. Bohlender [1], Steffen Pumple [1], Peter F. Zipfel [2,3], Karsten Häffner [4], Ralf Reski [1,5] & Eva L. Decker [1✉]

The complement system constitutes the innate defense against pathogens. Its dysregulation leads to diseases and is a critical determinant in many viral infections, e.g., COVID-19. Factor H (FH) is the main regulator of the alternative pathway of complement activation and could be a therapy to restore homeostasis. However, recombinant FH is not available. Engineered FH versions may be alternative therapeutics. Here, we designed a synthetic protein, MFHR13, as a multitarget complement regulator. It combines the dimerization and C5-regulatory domains of human FH-related protein 1 (FHR1) with the C3-regulatory and cell surface recognition domains of human FH, including SCR 13. In summary, the fusion protein MFHR13 comprises SCRs $FHR1_{1-2}$:$FH_{1-4}$:$FH_{13}$:$FH_{19-20}$. It protects sheep erythrocytes from complement attack exhibiting 26 and 4-fold the regulatory activity of eculizumab and human FH, respectively. Furthermore, we demonstrate that MFHR13 and FHR1 bind to all proteins forming the membrane attack complex, which contributes to the mechanistic understanding of FHR1. We consider MFHR13 a promising candidate as therapeutic for complement-associated diseases.

[1] Plant Biotechnology, Faculty of Biology, University of Freiburg, Freiburg, Germany. [2] Department of Infection Biology, Leibniz Institute for Natural Product Research and Infection Biology, Jena, Germany. [3] Institute of Microbiology, Friedrich Schiller University, Jena, Germany. [4] Faculty of Medicine, Department of Internal Medicine IV, Medical Center, University of Freiburg, Freiburg, Germany. [5] Signalling Research Centres BIOSS and CIBSS, University of Freiburg, Freiburg, Germany. ✉email: eva.decker@biologie.uni-freiburg.de

The complement is a fundamental part of the human immune system and constitutes the innate defense against infection agents. It consists of ~50 plasma and membrane-bound proteins forming a surveillance network, whose core function is the recognition and destruction of microbial invaders[1]. Complement activation can occur by the classical (CP), lectin (LP), or alternative (AP) pathways, which converge at complement component C3 activation and end up in membrane attack complex (MAC) formation, triggering lysis of invading pathogens and inflammation[2].

The AP contributes to up to 80% of the overall complement activation[3] and is spontaneously activated by hydrolysis of C3 to $C3(H_2O)$, with C3b-like activity. $C3(H_2O)$, together with the factor B (FB) fragment Bb, builds the initial C3 convertase ($C3(H_2O)Bb$) in fluid phase, which cleaves C3 to C3a and C3b. C3b mediates surface opsonization and amplifies complement activation, by building further C3 convertases (C3bBb) together with FB and FD. In addition, the AP acts as an amplification mechanism, even when complement was activated by the CP or LP[4]. As a consequence of excess C3b, C5 is activated, either by cleavage into C5a and C5b, or without proteolytic cleavage at very high densities of C3b on target surfaces, leading to C5b-like activated C5[5]. While C5b or C5b-like activated C5 bind to C6, C7, C8, and C9 leading to MAC formation (also called terminal complement complex (TCC)) and cell destruction, C3a and C5a are anaphylatoxins that trigger cell recruitment and inflammation[6] (Fig. 1a).

As complement activation can also damage intact body cells, the activation of the system is tightly controlled, especially by factor H (FH). FH is a 155 kDa glycoprotein, consisting of 20 globular domains, the short consensus repeats (SCR). The SCRs 1–4 (further named $FH_{1-4}$) act as a cofactor for factor I-mediated proteolytic degradation of C3b to inactive iC3b (which is subsequently degraded to C3c and C3dg and finally the latter to C3d), and accelerate the dissociation of C3 convertases in serum and on host cell surfaces[7,8]. FH binds to sialic acids and glycosaminoglycans on healthy host cells, protecting them from complement attack. The cell-surface-binding domains are mainly located on SCRs $FH_7$ and $FH_{20}$. The contribution of $FH_{13}$ has been a matter of discussion[9–11]. Recently, a heparin-binding site within fragment $FH_{11-13}$ was reported with an affinity constant around 17 μM, 4 and 14-fold weaker than that for $FH_{20}$ and $FH_7$, respectively, measured by Surface Plasmon Resonance (SPR)[12]. $FH_{13}$ has special structural characteristics which indicates functional importance. For instance, it is the smallest domain in FH with a positive net charge (+5), similar to $FH_7$, and contains the longest linkers in FH[11]. FH binds also through $FH_{19-20}$ to C3d, which contains the C3b-thioester domain (TED) that attaches to cell surfaces[13,14].

Five factor H-related proteins (FHRs) fine-tune FH-regulatory activity. The C-terminal region of FH ($FH_{19-20}$) is highly conserved in all FHRs, while none of them contains regions homologous to $FH_{1-4}$[8]. Of particular interest is FHR1, a regulator of C5 activation, which inhibits the last steps of the complement cascade (terminal pathway) and the MAC formation[13,15].

Mutations in FH, FHRs, or other complement-related genes are associated with diseases such as age-related macular degeneration (AMD), atypical hemolytic uremic syndrome (aHUS), C3 glomerulopathies (C3G), and paroxysmal nocturnal hemoglobinuria (PNH)[8,16]. However, protective haplotypes have also been identified; one of them is the polymorphism $FH^{V62I}$ (rs800292). Individuals carrying the variant I62 are less prone to complement-dysregulation diseases like aHUS, C3G, AMD, or dengue hemorrhagic fever[17,18]. Furthermore, $FH^{I62}$ showed

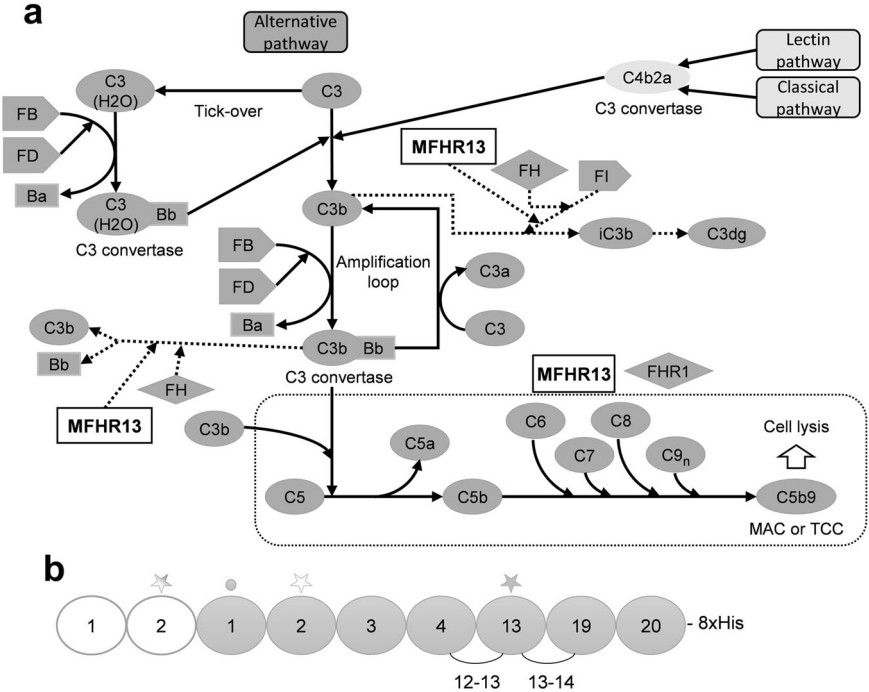

**Fig. 1 Schematic representation of complement activation and amplification by the alternative pathway (AP) and the mechanism of action of MFHR13. a** MFHR13 is a regulator of the AP activation at the level of C3 and C5. The complement is activated by three pathways which converge in the formation of C3 convertases and lead to the assembly of C5b-9, called membrane attack complex (MAC), able to induce lysis and cell death. The AP is activated by spontaneous hydrolysis of C3 and acts as an amplification loop for the cleavage of C3, even when complement was activated by the classical or lectin pathways. Dotted lines and box represent sites of downregulation of AP including the sites where MFHR13 act as regulator in the cascade. **b** MFHR13 consists of SCRs FHR$_{1-2}$ (white) and 7 SCRs of FH (gray) fused by natural linkers. The gray and white stars indicate glycosylation sites and a deamidated glycosylation site, respectively. The gray dot indicates the position of the polymorphism V62I.

increased binding affinity for C3b and enhanced cofactor activity[19].

Although the complement system should protect the body against viral infections, its over-activation has been associated with viral pathogenesis, e.g., in hepatitis C[20], dengue virus[21], and coronavirus infections[22,23]. Higher levels of C3a and C5a were detected in sera of patients with severe dengue[21] and deposition of complement proteins on hepatocytes was associated with liver damage in fatal cases[24]. Recently, the role of complement over-activation in SARS-CoV pathogenesis has been proved. C3 deposition was found in the lungs of SARS-CoV infected mice, while C3 KO mice suffered less from respiratory dysfunction[23]. Likewise, C5a accumulation and C3 deposition were observed in lung biopsy samples from COVID-19 patients[22], and enhanced activation of the alternative pathway was associated with a severe outcome of the disease[25]. Furthermore, C3a and C5a induce inflammation and are important in initiating the "cytokine storm", contributing to acute lung injury in COVID-19.

Therefore, anti-complement drugs might be an effective therapy to avoid severe inflammatory response[26,27]. Indeed, different pharmacological complement inhibitors, such as C1 esterase inhibitor[28], anti-C5 antibodies eculizumab, and ravulizumab[26,29], C3 inhibitor AMY-101[26], and anti-C5a antibody IFX-1[30] (the last two still in clinical trials) have been tested to treat severe cases of COVID-19, with promising results. However, the efficacy of these treatments and the best point of the complement activation cascade to target inhibition in COVID-19 patients still needs to be assessed. Furthermore, contrary to spike proteins from a common human coronavirus (HCoV-OC43), SARS-CoV-2 spike proteins were shown to activate the complement on cell surfaces mainly through the AP. Addition of FH protected cells from spike proteins-induced complement attack[31], which suggest that complement therapeutics based on FH activity might be an important alternative.

Although huge efforts are being undertaken to develop complement therapeutics, most are still in clinical development. Anti-C5 antibodies (eculizumab and ravulizumab) are efficient complement therapeutics and the only approved drugs to treat aHUS[32,33]. However, these compounds are not as effective in many patients suffering from C3G and would not be beneficial to treat severe dengue, because they do not prevent the cleavage of C3[21]. Thus, treatments also regulating the complement at the level of C3 are necessary. In general, anti-C5 antibodies, which block the terminal pathway and complement activity, are associated with a higher susceptibility to infections and are among the most expensive pharmaceuticals in the world[34].

It is therefore essential to generate new alternative therapeutic agents to treat diseases associated with complement dysregulation using approaches to control rather than block complement activation. FH is the physiological regulator on the level of C3 and therefore, FH-based therapies could restore homeostasis. However, due to the complexity of this molecule, it is desirable to produce smaller proteins with higher overall regulatory activity. Different fusion proteins including FH active domains have been developed [35-37]. The synthetic multitarget complement regulator MFHR1 combines the dimerization and C5-regulatory domains of FHR1 with the C3-regulatory and cell surface recognition domains of FH to regulate the activation of the complement in the proximal and the terminal pathways. This fusion protein (FHR$_{1-2}$:FH$_{1-4}$:FH$_{19-20}$) exhibited a higher overall regulatory activity in vitro compared to FH or miniFH (FH$_{1-4}$:FH$_{19-20}$) and prevented AP deregulation in models of aHUS and C3G[35].

The moss Physcomitrella (*Physcomitrium patens*) is a suitable production host for pharmaceutically interesting complex proteins, as demonstrated by the successful completion of the clinical trial phase I of α-galactosidase (Repleva AGAL; eleva GmbH) to treat Fabry disease[38]. MFHR1 and FH have been produced in moss with full in vitro regulatory activity[39-41].

Physcomitrella's characteristics include genetic engineering via precise gene targeting, growth as a differentiated tissue in a low-cost inorganic liquid medium, long-term genetic stability, industry-scale production in photo-bioreactors (500 L), homogenous glycosylation profile, high batch-to-batch stability and glycoengineering for improved pharmacokinetics and pharmacodynamics of the biopharmaceutical[38,42,43].

Here, we designed MFHR13 (FHR$_{1-2}$:FH$_{1-4}$:FH$_{13}$:FH$_{19-20}$, Fig. 1b) as a novel multitarget complement regulator produced in the GMP-compliant moss production platform. MFHR13 includes the variant I62 of FH, which we characterized to induce a higher binding to C3b and cofactor activity. After structure assessment by in silico modeling, we introduced the SCR FH$_{13}$, which includes an *N*-glycosylation site for higher protein stability[44], and contributes to increased flexibility of the molecule and cell surface recognition. MFHR13 was able to protect erythrocytes from complement attack in vitro much more efficiently than C5-binding antibodies, FH or MFHR1 (MFHR1$^{V62}$). Moreover, we could demonstrate that MFHR13, as well as its parental protein FHR1, are able to bind not only C5 or C5b6, but also C6, C7, C8, and C9, providing mechanistic insights into the role of FHR1 as a regulator of the complement system. We propose MFHR13 as a promising future biopharmaceutical to treat complement-associated diseases.

## Results

The protective polymorphism V62I is located in the regulatory region of FH, which is involved in C3b-binding and cofactor activity (CA). To confirm the suitability of the I62 variant as part of an improved complement regulator, we tested the influence of this single amino acid exchange on C3b binding and CA of MFHR1$^{I62}$ compared to MFHR1$^{V62}$, both produced in Physcomitrella.

**MFHR1$^{I62}$ was successfully produced in Physcomitrella.** MFHR1$^{V62}$ was obtained from the moss line P1 (IMSC no.: 40838)[41], and moss lines for the production of MFHR1$^{I62}$ were established. For this, the Δxt/ft moss parental line was used[45]. Recombinant proteins produced in this line display *N*-glycans lacking plant-specific fucoses and xyloses, which might trigger an immune response in patients. After transfection with pAct5-MFHR1$^{I62}$ and selection, suspension cultures were established for all surviving plants and screened for MFHR1$^{I62}$ production via ELISA. 70% of the lines produced MFHR1$^{I62}$ in different concentrations (Supplementary Fig. 1). Four of the best producing lines were compared regarding growth and protein productivity at shake-flask scale during 26 days (Supplementary Fig. 2). The best producing line, N-179 (IMSC no.: 40839), was scaled up to a 5 L stirred bioreactor, where a productivity of 170 μg MFHR1$^{I62}$/g FW was achieved (Supplementary Fig. 3). For structure and activity characterization, MFHR1$^{I62}$ was purified from 8-days-old moss tissue via nickel affinity chromatography followed by anion exchange chromatography. Approximately 500 μg MFHR1$^{I62}$/mL were obtained after purification and concentration.

**The MFHR1$^{I62}$ variant has higher C3b binding and cofactor activity**. To analyze the influence of the polymorphism V62I on protein function, we assessed the binding of MFHR1$^{V62}$ and MFHR1$^{I62}$ to C3b and C5, respectively, via ELISA. As expected, there was no difference between both variants in C5 binding ($P = 0.9328$, Fig. 2a), while MFHR1$^{I62}$ bound to immobilized C3b significantly better than MFHR1$^{V62}$ ($P = 0.0437$ Fig. 2b, Supplementary Table 2). The absorbance values from one representative experiment are shown in Supplementary Fig. 7.

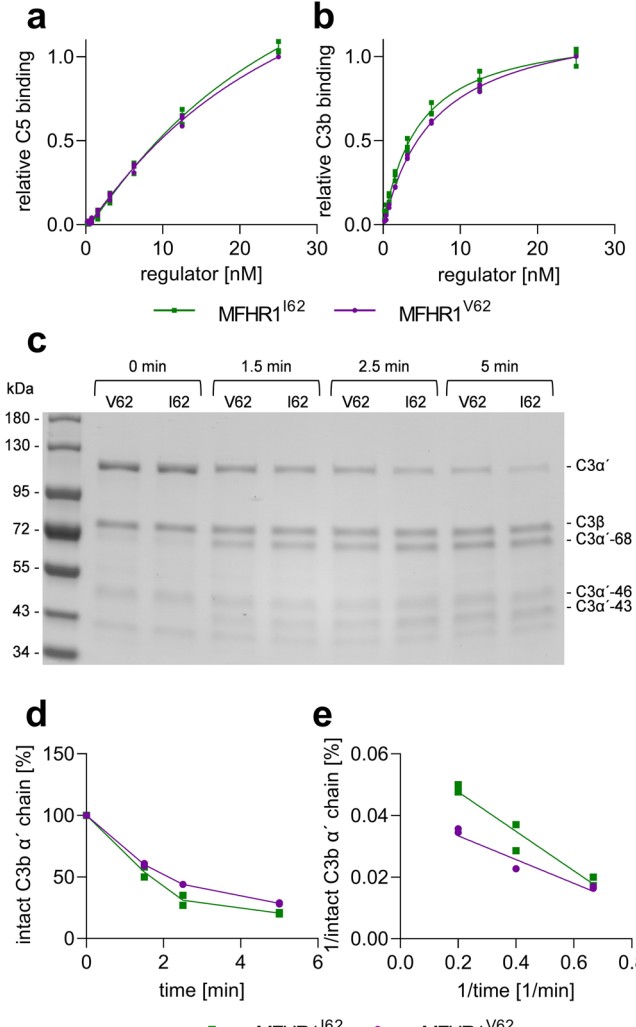

**Fig. 2 MFHR1[I62] displays better C3b binding and cofactor activity than MFHR1[V62]. a** Binding to immobilized C5 was evaluated by ELISA with increasing concentrations of MFHR1[I62] and MFHR1[V62]. Nonlinear regression using the sigmoidal equation and logarithmic transformed data showed no significant differences between both variants binding to C5 ($P = 0.9328$). Data represent mean values ± SD from 3 independent experiments. **b** Binding to immobilized C3b was evaluated by ELISA with increasing concentrations of MFHR1[I62] and MFHR1[V62]. Nonlinear regression using the sigmoidal equation (dose–response curve, 4 parameters) showed significant differences between both variants binding to C3b ($P = 0.0437$). Data from one representative experiment are shown in Supplementary Fig. 7. **c** Cofactor activity of FI-mediated proteolytic cleavage of C3b α′-chain in the presence of MFHR1[I62] or MFHR1[V62] was visualized by SDS-PAGE and Coomassie staining. One representative experiment is shown. **d** Densitometric analysis of C3b cleavage. The amount of intact α′-chain was normalized for each sample with the β-chain and set to 100% intact C3b α′-chain at time zero. Two independent experiments were included in the analysis. **e** Double reciprocal plot of the percentage of intact C3b α′-chain. Linear regression analysis indicated significant differences between the MFHR1 variants ($P = 0.0302$).

CA of both variants was measured in fluid phase. Degradation of C3b is characterized by the cleavage of the C3b α′-chain to fragments C3b α′-68, -46, and -43, while C3b β-chain is not cleaved (Fig. 2c). MFHR1[I62] displayed slightly better CA than MFHR1[V62] (Fig. 2d). Double reciprocal plot and linear regression analysis showed significant differences between them ($P = 0.0302$,

Fig. 2e). Due to the positive influence of I62 in the regulatory activity, we included this variant in MFHR13.

**Comparative protein structure modeling approach of MFHR13.** To provide a potent complement biopharmaceutical, structural features that promote the efficacy of the product have to be taken into account: (1) Improving flexibility of the molecule by increasing the distance between FH4 and FH19, which will be important to enhance the biological activity of engineered versions of FH[46]. (2) Improving glycosylation, since it plays an important role in the stability, half-life in the circulation, efficacy, and safety of biopharmaceuticals[44]. MFHR1 has two glycosylation sites. The first site in FHR1₂ is partially occupied and the second site located in FH4 is deamidated and thus not glycosylated[41,47], resulting in a high portion of non-glycosylated product. Therefore, we included an additional SCR with an *N*-glycosylation site to MFHR1[I62] between FH4 and FH19. The glycosylation sites in FH are found in SCRs 9, 12, 13, 14, 15, 17, and 18. An interesting domain was FH13 due to the high amount of basic amino acid residues and its electrostatic properties, which suggests it might contain a polyanion (glycosaminoglycans or sialic acid) binding site, thus potentially increasing the host cell surface recognition.

A comparative protein structure modeling approach (Modeller 9.19) was carried out to evaluate the structure of the fusion protein FHR1₁₋₂:FH1₋₄:FH13:FH19₋₂₀ and if the C3b binding sites are accessible.

To build the model we used crystal structures retrieved from the PDB database corresponding to FHR1₁₋₂, FH1₋₄ +C3b, FH12₋₁₃, FH19₋₂₀, and FH19₋₂₀ + C3d. Furthermore, FH1114 (Model SAXS, FH11₋₁₄) was used to orient FH13, since FH14 shares about 30% sequence identity with FH19. Out of 400 models generated, we selected the one with the lowest DOPE score and assessed its quality using the web-based tools ProSa and PROCHECK. The overall quality of the model was acceptable and in the range of all available structures on PDB. According to the local model quality, where the energy is plotted as a function of residues' position in the sequence, problematic parts of the model (characterized by positive energy values) were located in some linkers, such as the linker between FH4 and FH13. However, this is also observed in linkers of the native proteins of this family (Supplementary Fig. 4a, b). Moreover, according to the Ramachandran plot, 97.6% of amino acid residues are in favored regions, 1.3% in allowed regions, and only 1.1% in outlier regions (Supplementary Fig. 4c). Our model indicates that the introduction of FH13 may not affect biological activity and might confer higher flexibility to the molecule (Fig. 3a). Finally, the MFHR13 model was superimposed with the structures 2WII and 2XQW, which correspond to FH1₋₄ bound to C3b and FH19₋₂₀ to C3dg, respectively. The natural linkers are flexible enough to allow FH13 to fold back over in such a way that allows FH1₋₄ and FH19 to interact simultaneously with one C3b molecule (Fig. 3b). Furthermore, there is a striking electropositive patch extending over one face of FH13 to FH20 (Fig. 3c), which might enhance the affinity to host cell surfaces.

**Moss-produced MFHR13 is correctly synthesized and efficiently glycosylated.** For the production of MFHR13 (FHR1₁₋₂:FH1₋₄:FH13:FH19₋₂₀), the Δxt/ft moss line was transformed with the pAct5-MFHR13 construct. Selection and screening of plants producing MFHR13 were performed following the same procedure as described above for MFHR1[I62]. From 98 surviving plants, 45% produced MFHR13 in different levels as screened by ELISA (Supplementary Fig. 5). For the best producing line N13-49 (IMSC no.: 40840) around 70 μg MFHR13/g FW were achieved in the 5 L bioreactor cultivation, which correspond to 7 mg MFHR13 after 8 days (Supplementary Fig. 6).

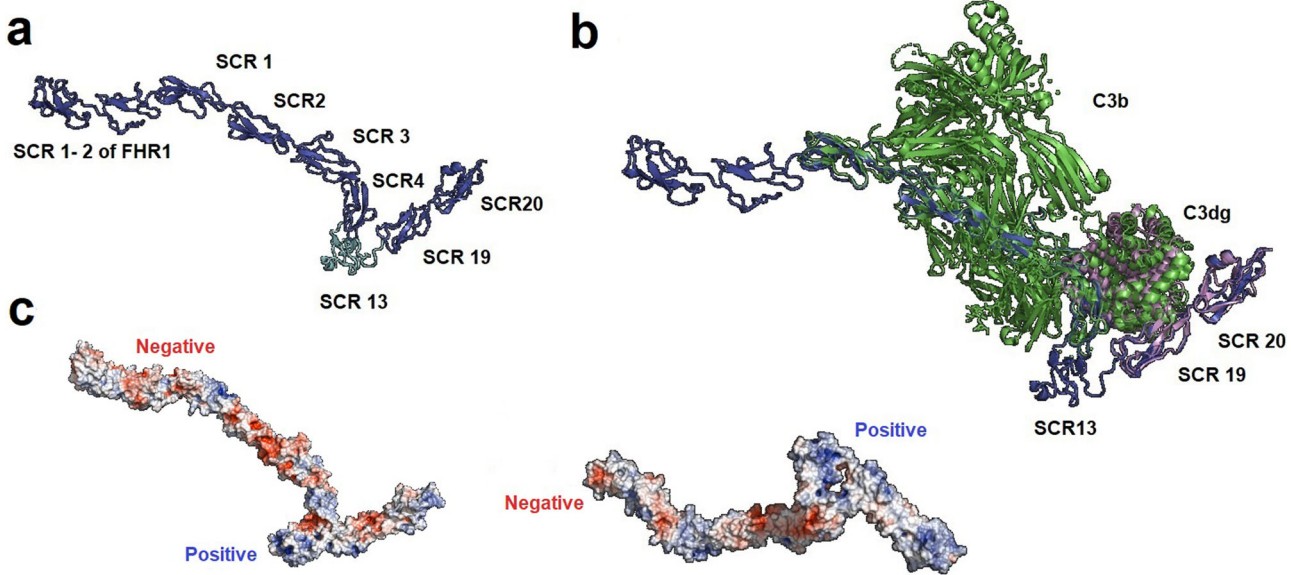

**Fig. 3 Comparative protein structure modeling of the fusion protein MFHR13 (FHR1_{1-2}:FH_{1-4}:FH_{13}:FH_{19-20}). a** Model of MFHR13 built in Modeller 9.19. **b** Superimposition of the model MFHR13 (blue) with C3b (green) and C3dg (purple), respectively. Root-mean-square deviation of atomic positions (RMS) between the structure C3b: FH_{1-4} (PDB 2WII) and modeled MFHR13 is 1.345. **c** Electrostatic potential of MFHR13. Electronegative residues are displayed in red, electropositive in blue. The electropositive patch extending over one face of FH_{13} to FH_{20} is on the right-hand side. The electrostatic potential was determined using Adaptive Poisson-Boltzmann Solver included in PyMOL 2.0.

Approximately 200 μg MFHR13/mL were obtained after purification (Fig. 4a). The concentration was measured by ELISA and the correct ratios between MFHR13, MFHR1^{I62}, and MFHR1^{V62} were further confirmed by semi-quantitative western blot (Fig. 4b).

MFHR13 migrated at ~70 kDa and the integrity of the protein was confirmed by western blot and mass spectrometry (MS) (Fig. 4a–c). Via MS, MFHR13 was identified with a sequence coverage of 64%. The correct introduction of SCR13 of FH, together with its linkers, and the presence of isoleucine at position 193 were confirmed (Fig. 4c). The glycosylation site (NIS) located in the FHR1_2 in MFHR13 was around 65% occupied with the pattern described in Fig. 4d. As already described for moss-produced MFHR1 and human FH, the glycosylation site located on FH_4 in MFHR13 was deamidated and not glycosylated. The new glycosylation site (NCS) introduced within FH_{13} was fully glycosylated, as we did not detect the unglycosylated peptide. Around 85% of all glycans comprised mature complex-type glycans (GnGn, GnAF, AAFF) and only 15% of immature glycans with terminal mannose (Fig. 4d).

**MFHR13 exhibits enhanced heparin-binding.** One important feature of complement regulators is the ability to bind to poly-anions such as glycosaminoglycans and sialic acids on host cell surfaces and protect them from complement attack. The ability of MFHR13 to bind to the polyanion heparin was evaluated by an ELISA-based method, and compared to MFHR1^{V62}, MFHR1^{I62}, and hFH, respectively. As expected, there were no differences in heparin-binding between MFHR1^{V62} and MFHR1^{I62}. In contrast, MFHR13 bound to heparin twice as strong as MFHR1^{V62} and MFHR1^{I62} ($P < 0.0001$) (Fig. 5a, Supplementary Table 3, Supplementary Fig. 8a). This can be attributed to the presence of FH_{13}. Heparin-bound hFH was only detected at the highest concentration tested (25 nM), indicating weaker binding to heparin than MFHR13 (Fig. 5b, $P < 0.0001$).

**MFHR13 regulates complement activation at the level of C3.** Binding of MFHR13 to immobilized C3b was evaluated via

ELISA in comparison to MFHR1^{I62}, to evaluate the influence of FH_{13} on this activity. We did not find significant differences between binding of both proteins to C3b (Fig. 6a; $P = 0.8426$, Supplementary Table 4, Supplementary Fig. 8b).

MFHR13 is able to support the FI-mediated cleavage of C3b α′-chain. MFHR13 showed similar fluid-phase CA in independent experiments as compared to MFHR1^{I62}, however, hFH had a higher CA than both synthetic regulators. (Fig. 6b, c) ($P = 0.0025$ and 0.0040 for MFHR13 and MFHR1^{I62}, respectively).

The ability of the regulators to dissociate preformed C3 convertases, known as decay acceleration activity (DAA), was evaluated in an ELISA-based method. Increasing equimolar amounts of the regulators MFHR13, MFHR1^{I62}, MFHR1^{V62}, or hFH, respectively, were incubated with C3 convertase (C3bBb) and Bb fragment displacement was detected. The DAA of MFHR13, with an IC_{50} of 2.7 nM, was similar to that of hFH (IC_{50} 3.8 nM) ($P = 0.2022$), but it significantly exceeded the activity of MFHR1^{I62} and MFHR1^{V62} (IC_{50} 7.8 and 9.4 nM, respectively) ($P < 0.0001$). (Fig. 6d, Supplementary Table 5).

**MFHR13 blocks the terminal pathway of complement**
*FHR1 and MFHR13 bind to C5, C5b6, C6, C7, C8, and C9.* FHR1 and MFHR1 are able to interact with C5[35]. Binding of MFHR13 to immobilized C5 was evaluated via ELISA in comparison to MFHR1^{I62}. There was no significant difference after fitting the data with 4PL nonlinear regression model (Fig. 7a; $P = 0.2565$, Supplementary Fig. 8c).

Further, we tested the binding of MFHR13, the MFHR1 variants, FHR1 and hFH to C5b6, and the other proteins involved in MAC formation. Immobilized C5b6, C6, C7, C8 and C9, and C3b and C5 as positive controls, were incubated with 25 nM of MFHR13 or MFHR1 variants (8x His-tag) or 50 nM FHR1 (6x His-tag) and bound protein was detected with anti-His-tag antibodies. Due to the differences in the amount of histidine in the tag, FHR1 cannot be quantitatively compared to the moss-produced regulators. Due to the lack of His-tag, binding of hFH to the complement proteins was evaluated separately using anti-FH polyclonal antibodies and compared with MFHR1^{V62}. As

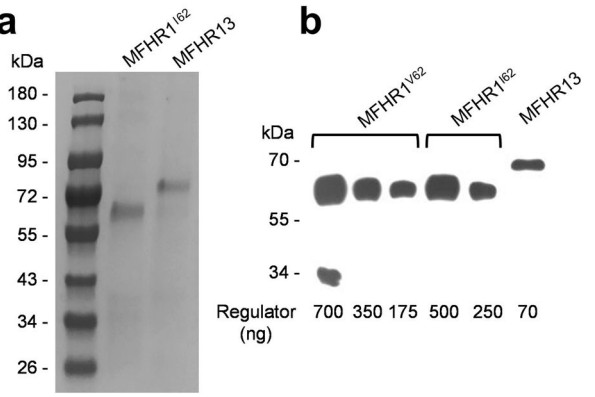

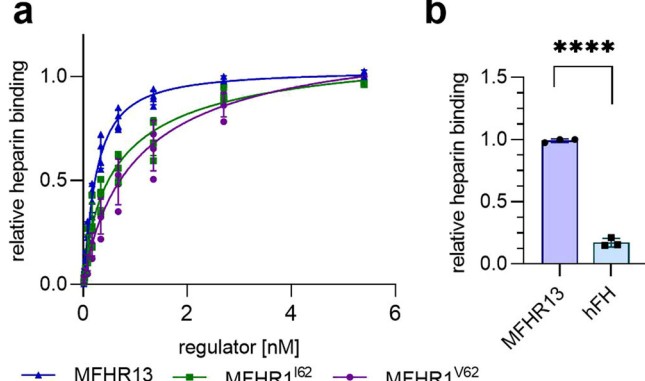

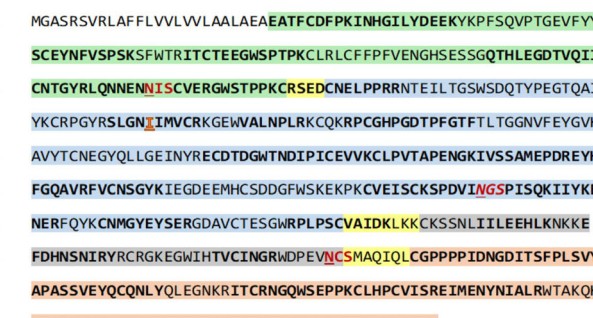

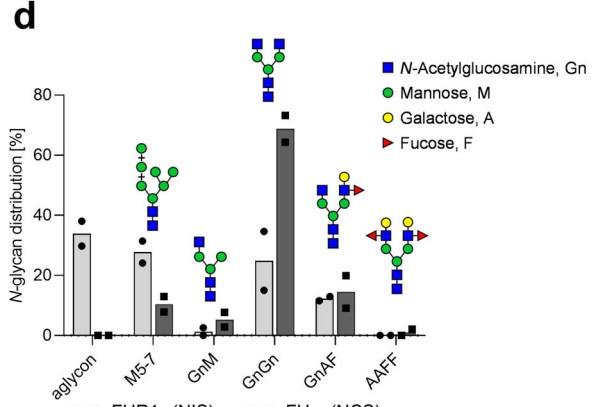

**Fig. 5 MFHR13 binds better to heparin than MFHR1^I62, MFHR1^V62, or hFH. a** Heparin binding was evaluated by ELISA. Nonlinear regression using the sigmoidal equation (dose–response curve, 4 parameters) and comparison of $IC_{50}$ showed significant differences between MFHR13, MFHR1^I62, and MFHR1^V62 binding to heparin ($P < 0.0001$ both comparisons). Data represent mean values ± SD from 4 independent experiments. The blank (BSA 2%) was subtracted from all the samples. Data from one representative experiment are shown in Supplementary Fig. 8a. **b** MFHR13 binds significantly better to heparin than hFH ($P < 0.0001$, unpaired $t$-test). The binding was measured by ELISA using 25 nM of hFH or MFHR13, respectively.

**Fig. 4 MFHR13 structure and sequence characterization. a** Coomassie staining of purified MFHR1^I62 and MFHR13 via Ni-affinity and anion exchange chromatography. Samples were separated on SDS-PAGE under reducing conditions. **b** Semi-quantitative western blot of MFHR1^V62, MFHR1^I62, and MFHR13 under reducing conditions and detected with anti-His-tag antibodies. A calibration curve was obtained with MFHR1^V62 and densities of the western blot signal were compared with the concentrations obtained by ELISA (included under each lane). One representative experiment is shown. **c** MFHR13 sequence. FHR1_{1-2} are shown in green, FH_{1-4}, FH_{13}, FH_{19-20} are shown in blue, gray and light red, respectively, and the linkers are shown in light yellow. The glycosylation sites are highlighted in red and the deamidated site in italics. The peptides identified by MS are shown in bold. **d** Relative quantification of glycopeptides based on MS. MFHR13 was produced in stirred tank bioreactor (5 L) and extracted at day 8. NIS: glycosylation site in the FHR1_2 in MFHR13. NCS: glycosylation site in the FH_{13} in MFHR13. Mean values from two technical replicates with standard deviation are shown.

expected, FH bound only to C3b (Fig. 7b, Supplementary Fig. 9). In contrast, MFHR13, MFHR1 variants and FHR1 bound to all complement proteins involved in MAC formation (Fig. 7c, Supplementary Fig. 9). Here, we proved that FHR1 binds not only to C3b, C5, and C5b6 but also to C6, C7, C8, and C9, which contributes to the mechanistic understanding of FHR1 as complement regulator.

ELISA is a widely used technique to observe protein interactions. However, to validate the results obtained via ELISA, we evaluated the binding of MFHR13 with selected components of the MAC, C5, C7, and C9, in equilibrium by Microscale Thermophoresis (MST), an immobilization-free technique. We included C3d as a positive control. Using this technique, we proved that the interaction between MFHR13 and the selected MAC components occurs with an equilibrium dissociation constant ($K_D$) in the low micromolar range. However, it was not possible to reach saturation of the binding curve, which prevents the estimation of an accurate $K_D$ (Fig. 7d).

*MFHR13 prevents hemolysis by blocking MAC formation from C5b6.* The C5-regulatory activity of FHR1 is still being discussed because of difficulties in separating C3 and C5-regulatory activities in vivo. FHR1 and MFHR1 were shown to bind C5b6 and prevent MAC formation in vitro[13,35]. Here, we evaluated this activity of moss-made MFHR13, using a serum-free approach and compared their activity to FHR1, hFH, and eculizumab. For this, the complex C5b6 was incubated with the regulators (1 µM) followed by the addition of sheep erythrocytes and the terminal complement proteins C7, C8, and C9. Inhibition of C5b-9 formation was indirectly measured by hemolysis. The surface of sheep erythrocytes is rich in sialic acid and a good model for host-cell surfaces to test regulation of complement attack.

MFHR13, similar to FHR1, protected sheep erythrocytes from complement attack, reducing cell lysis by ~70%, while hFH, eculizumab, BSA, and purified extract from the parental line (Δxt/ft) lacked such positive effects (Fig. 7e).

The terminal pathway is initiated upon cleavage of C5 to C5b, or independently from convertases when C5 binds to densely C3b-opsonized surfaces and acquires a C5b-like conformation[5]. Subsequently, MAC formation begins with the binding of C5b, or C5b-like activated C5, to C6. Here, we evaluated the ability of MFHR13 to inhibit the C5 priming on opsonized sheep erythrocytes. Human FH did not differ significantly from the

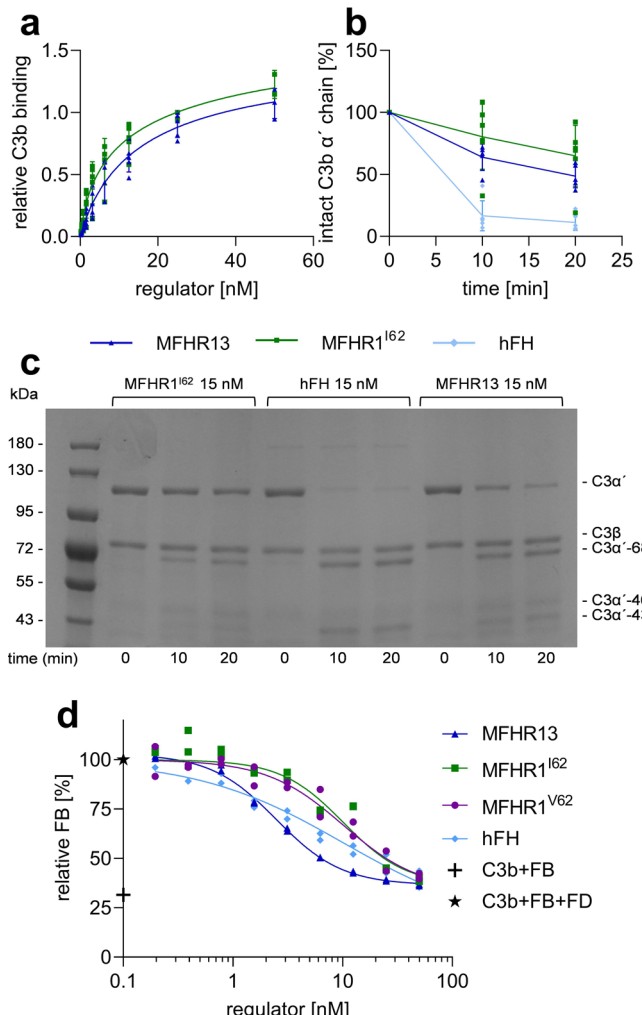

**Fig. 6 MFHR13 binds to C3b and exhibits cofactor and decay acceleration activities in fluid phase. a** C3b binding was evaluated by ELISA. Nonlinear regression using the sigmoidal equation (dose–response curve, 4 parameters) and comparison of IC$_{50}$ showed no significant differences between MFHR13 and MFHR1$^{I62}$ binding to C3b ($P = 0.8426$). Data represent mean values ± SD from 5 independent experiments. Absorbance values from one representative experiment are shown in Supplementary Fig. 8b. **b** Cofactor activity of FI-mediated proteolytic cleavage of C3b α'-chain in the presence of MFHR1$^{I62}$, FH, or MFHR13, evaluated in increasing reaction times. Densitometric analysis of C3b cleavage. The amount of intact α'-chain was normalized for each sample with the β-chain and set to 100% intact C3b α'-chain at time zero. The data represent mean value from 5 or 6 independent experiments ± SD. **c** One representative experiment included in b is shown. C3b, FI, and MFHR1$^{I62}$, MFHR13 or FH were incubated for 10 and 20 min and C3b cleavage was visualized by SDS-PAGE and Coomassie staining. **d** MFHR13 displaced C3 convertases more efficiently than MFHR1$^{I62}$ and MFHR1$^{V62}$. C3bBb were assembled in C3b-coated microtiter plates with FB and FD. Complement regulators were added and the dissociation of the C3 convertases was detected by the amount of FB using polyclonal anti-FB antibodies. Absorbance of samples without regulator was set to 100% FB and samples without FD as the negative control. Nonlinear regression using the sigmoidal equation and comparison of IC$_{50}$ indicated significant differences in DAA between MFHR13, MFHR1$^{V62}$, and MFHR1$^{I62}$ ($P < 0.0001$ for each comparison) and not significant difference between MFHR13 and hFH ($P = 0.2022$). The data represent mean value from 2 independent measurements ± SD.

negative controls BSA or purified extract from Δxt/ft, and could barely inhibit hemolysis. Under these conditions, the sample treated with eculizumab supported hemolysis, which was unexpected and differs from what was reported before when higher concentrations of complement proteins and antibody have been used[5]. In contrast, MFHR13 (700 nM) reduced cell lysis by 60%, efficiently protecting sheep erythrocytes from convertase-independent C5 activation and MAC formation, while FHR1 reduced lysis by ~33% (Fig. 7f).

Interestingly, the hemolysis induced by C5b6-C9 was (~130%) higher than the one induced by convertase-independent activation of C5, even when the concentrations of complement proteins involved in the MAC formation were almost 20 times higher in the latter assay (Supplementary Fig. 10).

**MFHR13 exhibits efficient regulatory activity on activated AP**. The ability of MFHR13 to inhibit the formation of the C5b-9 complex (TCC) after activation of the AP with LPS in normal human serum (NHS) was evaluated by an ELISA-based assay. The inhibitory activity of MFHR13 was compared to MFHR1$^{I62}$, MFHR1$^{V62}$, hFH, and eculizumab. MFHR13 inhibited TCC formation very efficiently, exhibiting 37-fold and 2-fold the activity of hFH and eculizumab respectively (Fig. 8a; Table 1). MFHR13 inhibited slightly better than MFHR1$^{V62}$ and no significant difference was observed compared to MFHR1$^{I62}$.

**MFHR13 protects sheep erythrocytes from complement-mediated lysis**. Finally, we tested the ability of the moss-produced MFHR13 to protect sheep erythrocytes from complement-mediated lysis. The AP was activated in NHS depleted of FH by addition of MgEGTA. In this in vitro-model for host cell surfaces, MFHR1$^{V62}$ and MFHR1$^{I62}$ were as active as hFH. However, MFHR13 protected sheep erythrocytes from complement attack more efficiently than MFHR1$^{V62}$ (3.7-fold), MFHR1$^{I62}$ (2.9-fold), hFH (4.2-fold), and eculizumab (26-fold) (Fig. 8b, Table 1).

## Discussion

Complement dysregulation due to mutations in complement genes, autoantibodies against complement proteins or over-activation in response to certain viral infections leads to a long list of diseases. C3G, PNH, and aHUS are examples where complement is the main driver of the disease[8,16,48]. Although the development of drugs to regulate the complement response and restore homeostasis is of huge interest, only very few so far are available on the market, e.g., C1 esterase inhibitor and blocking antibodies against C5 (eculizumab, ravulizumab)[32]. FH is a physiological regulator of the complement system, but truncated versions of it, such as mini-FH (AMY201, Amyndas)[46], are being evaluated as they have been proposed as better therapeutics because of an enhanced overall regulatory activity compared to FH. Moreover, drugs able to regulate the activation cascade at several levels might be especially interesting to increase efficiency. MFHR1 combines the regulatory domains of FHR1 and FH, and exhibits a higher regulatory activity than mini-FH (FH$_{1-4}$:FH$_{19-20}$)[35]. Our studies led to the development of MFHR13 as a potential therapeutic for complement dysregulation.

For MFHR13 design, we first considered the polymorphism V62I. The substitution of valine by isoleucine within SCR1 of FH is located in the regulatory domain. FH$^{I62}$ is associated with lower risk for aHUS and C3G[49]. Although both amino acids are biochemically similar, and both FH variants can fold correctly, the substitution has structural consequences: The FH SCRs are tightly folded forming a compact hydrophobic core, where the residue 62 is buried, and the additional methylene group from isoleucine triggers a small

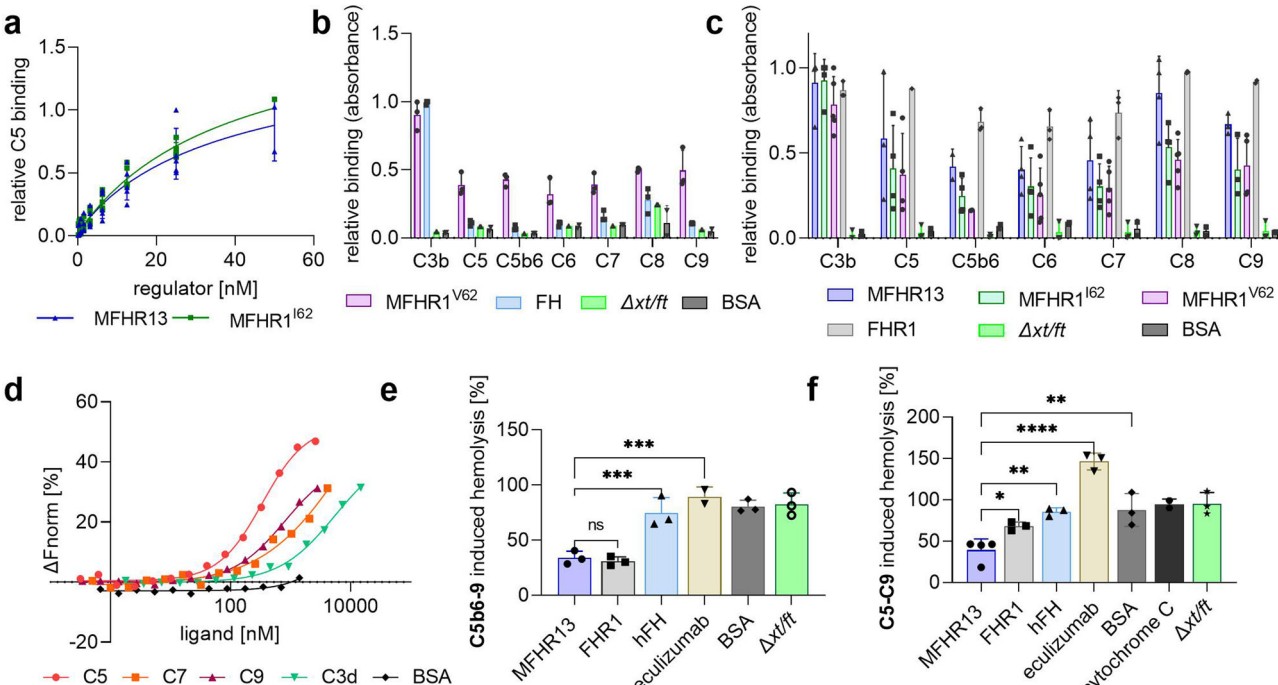

**Fig. 7 MFHR13 binds C5 and MAC components and blocks the terminal pathway. a** C5 binding was measured by ELISA. Nonlinear regression using the sigmoidal equation and comparison of $IC_{50}$ showed similar binding profiles for MFHR13 and MFHR1[I62] to C5 ($P = 0.2565$). Data represent mean values ± SD from 4 or 5 independent experiments. Absorbance values from one representative experiment are shown in Supplementary Fig. 8c. **b** MFHR1 binds to the individual TCC proteins; hFH does not. MFHR1 or hFH (25 nM) as well as the negative controls BSA and the purified extract from the parental line (Δxt/ft) were added to immobilized complement proteins in microtiter plates and detected by anti-FH polyclonal antibodies. Data represent mean values ± SD from 3 independent experiments. **c** MFHR13, MFHR1, and FHR1 bind to the terminal complement proteins involved in MAC formation. Regulators (25 or 50 nM) were added to immobilized terminal complement proteins and bound proteins were detected by anti-His-tag antibodies. Data represent mean values ± SD from 3 or 4 independent experiments. Absorbance values from one representative experiment from (**b**) and (**c**) are shown in Supplementary Fig. 9. **d** Binding of NT-647 RED-NHS-labeled MFHR13 to C3d, C5, C7, and C9 was evaluated by Microscale Thermophoresis (MST) in fluid phase. Normalized fluorescence (ΔFnorm) was plotted against the concentration of single complement components. **e** MFHR13 and FHR1 inhibit the formation of the MAC on sheep erythrocytes. The inhibition by MFHR13 and FHR1 was significantly better than hFH ($P = 0.0018$), eculizumab ($P < 0.0001$), BSA ($P = 0.0012$), or Δxt/ft. Hemolysis of sheep erythrocytes was induced with addition of C5b6, C7, C8, and C9 and release of hemoglobin was determined by measuring the absorbance at 405 nm. The average absorbance values without C5b6 and without C9 were subtracted from the positive control and samples. Lysis induced by TCC in absence of regulators was set to 100%. Data represent mean values ± SD from three independent experiments (one-way ANOVA, Bonferroni post hoc test). **f** MFHR13 (700 nM) inhibits MAC formation on C3b-opsonized sheep erythrocytes significantly better than FHR1 ($P = 0.0415$), hFH ($P = 0.0018$), eculizumab ($P < 0.0001$), BSA ($P = 0.0012$), cytochrome c ($P = 0.0007$), and Δxt/ft. Hemolysis of C3b-coated sheep erythrocytes was induced with a mix of C5-C9 and determined by absorbance of released hemoglobin at 405 nm. Lysis without regulators was set to 100%. The average absorbance values without C5 were subtracted from all samples. Data represent mean values ± SD from three independent experiments (one-way ANOVA, Bonferroni post hoc test). ****$P ≤ 0.0001$, ***$P ≤ 0.001$, **$P ≤ 0.01$, and *$P ≤ 0.05$, ns (no significant difference).

conformational rearrangement. In addition, the FH[I62] is slightly more thermally stable and rigid than the FH[V62] [50]. We studied the influence of this amino acid exchange on the activity of the synthetic complement regulator MFHR1. Binding to immobilized C3b and cofactor activity of MFHR1[I62] were significantly higher than that of MFHR1[V62]. Therefore, we included the more active variant I62 in MFHR13.

The importance of protein glycosylation on the stability, efficacy, and half-life of biopharmaceuticals in serum is well known[44]. Deglycosylated proteins were shown to be inactive in vivo[51]. MFHR1[V62] has a higher regulatory activity than hFH in vitro but was surpassed by hFH in vivo, which was attributed to a reduced half-life in the circulation[35]. MFHR1[V62] is only partially glycosylated, due to the nature of the N-glycosylation sites present in this fusion protein, which are not or only partially glycosylated in the physiological counterparts FH and FHR1, respectively[41]. To support protein stability and prolong half-life in serum we aimed to include an additional N-glycosylation site in the new MFHR. For this purpose, we searched for glycosylated SCRs from FH, which beyond achieving glycosylation without the

addition of artificial sequences would increase the distance between SCRs FH$_4$ and FH$_{19}$ in the regulator. This distance has been postulated before to be crucial for the flexibility and the activity of mini-FH[46].

We proposed the introduction of FH$_{13}$ with its natural linkers to generate the new molecule MFHR13. FH$_{13}$ harbors many striking characteristics: Apart from displaying an N-glycosylation site, it is the smallest SCR in FH with the most flexible linkers and one of the most electrostatic positive domains. It has an unusual distribution of charged groups with a net charge of +5, similar to FH$_7$ and FH$_{20}$[52]. We hypothesized that the host cell surface recognition could be enhanced by adding this SCR. A comparative protein structure modeling approach supported our hypothesis, since FH$_{13}$ was completely exposed with a positive electrostatic patch extending towards FH$_{20}$. This feature may increase the binding to cell surfaces due to electrostatic interactions and thus stabilize the cell surface-binding site in FH$_{20}$.

The production of this complement regulator was carried out in moss bioreactors, due to the advantages offered by this system. MS

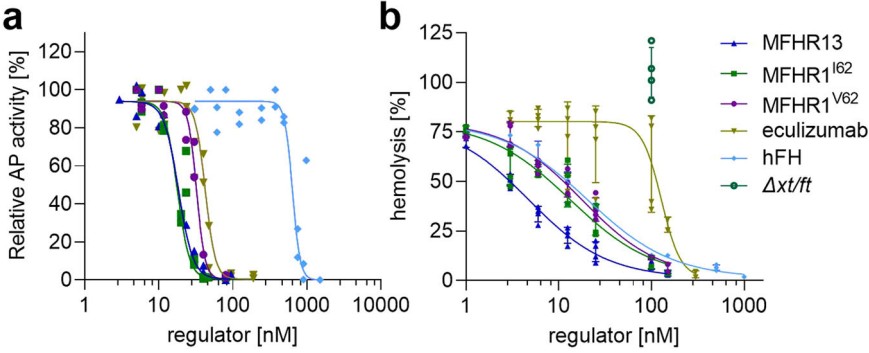

**Fig. 8 MFHR13 regulates overall AP activation and protects host-like surfaces. a** Regulation of AP activation. AP in normal human serum was activated in LPS coated wells. The inhibition achieved by addition of regulators was evaluated indirectly by quantifying C5b-9 complex formation by an ELISA-based method. AP activity was set to 100% for wells without regulators and wells with heat-inactivated serum were used as blank. Spontaneous activation was 42 ± 2.8%, measured in wells without LPS. Data represent mean values ± SD from 4 independent measurements. MFHR13 regulatory activity was significantly better than hFH ($P < 0.0001$, $F_{(DFn, DFd)} = 268.5\ (1, 34)$), and eculizumab ($P < 0.0001$, $F_{(DFn, DFd)} = 83.38\ (1, 25)$). **b** Increasing concentrations of MFHR13 protected sheep erythrocytes from human complement attack in FH-depleted serum significantly better than MFHR1^I62 ($P = 0.0003$, $F_{(DFn, Dfd)} = 16.02\ (1, 36)$), MFHR1^V62 ($P < 0.0001$, $F_{(DFn, Dfd)} = 54.99\ (1, 42)$), hFH ($P = 0.0001$, $F_{(DFn, Dfd)} = 18.73\ (1, 31)$), and eculizumab ($P < 0.0001$ $F_{(DFn, Dfd)} = 111.7\ (1, 33)$). Hemolysis was set to 100% for wells without complement regulators and the negative control without FH-depleted serum was used as blank. Data represent mean values ± SD from 4 independent measurements. *Δxt/ft* is a purified extract from the parental moss line used as negative control. DF degrees of freedom.

**Table 1 Overall AP regulatory activity evaluated by hemolysis and an ELISA-based method: Best fit IC$_{50}$ calculated by 4PL nonlinear regression model.**

| Assay | MFHR13 | MFHR1^I62 | MFHR1^V62 | hFH | Eculizumab |
|---|---|---|---|---|---|
| *AP ELISA* | | | | | |
| IC$_{50}$ (nM) | 17.8 | 19.3 | 33.1 | 665 | 38.2 |
| 95% CI | 16.9–18.9 | 16.1–22.5 | 29.4–37.4 | 477.2–893.3 | 34–42.9 |
| $R^2$ | 0.99 | 0.95 | 0.95 | 0.78 | 0.96 |
| *Hemolysis* | | | | | |
| IC$_{50}$ (nM) | 4.9 | 14.3 | 18.4 | 21.3 | 128.8 |
| 95% CI | 3.7–7.2 | 10.8–21.7 | 14.3–27.2 | 14.8–37.0 | 115.5–152.6 |
| $R^2$ | 0.96 | 0.89 | 0.96 | 0.98 | 0.70 |

*CI* confidence interval, $R^2$ goodness of fit.

analysis revealed that the FH$_{13}$-*N*-glycosylation site was fully occupied with more than 85% mature complex-type *N*-glycans, predominantly GnGn and around 15% carrying the Lewis A epitope. Although this structure is not desired, the gene responsible for the generation of this structure was identified in moss and can be eliminated by gene targeting[53]. In order to increase the half-life in the circulation, *N*-glycans still need to be terminally sialylated to avoid rapid clearance from circulation[54]. Stable protein sialylation is feasible in Physcomitrella[55], although the efficiency should still be improved.

In analogy to FH, MFHR13 should protect host cells against complement attack, directing the complement regulatory activity to the cell surface by interaction with polyanions. Therefore, we tested the binding to heparin of MFHR13 compared to MFHR1^I62, our control lacking FH$_{13}$. MFHR13 binds significantly better to heparin than MFHR1^I62, indicating that FH$_{13}$ retains the function of the polyanion binding site described recently for FH$_{11-13}$[12]. Moreover, MFHR13 binds to heparin much better than hFH.

Next, we tested the regulatory activity of MFHR13 at the level of C3. From our protein structure modeling, we expected an enhanced binding of MFHR13 to C3b due to increased flexibility of the molecule. In addition, a weak C3b binding site localized to FH$_{13-15}$ was described recently[12]. MFHR13 binds efficiently to C3b, however, we could not detect significant differences to MFHR1^I62. This indicates that FH$_{13}$ alone is not enough to build the C3b-binding site described for FH$_{13-15}$.

MFHR13 displayed similar cofactor activity in fluid phase to MFHR1^I62, while hFH outperformed both, possibly due to small steric hindrances generated by the dimerization domains present in FHR1$_{1-2}$. MFHR13 had a similar decay acceleration activity compared to hFH and significantly higher than MFHR1^I62 or MFHR1^V62. Our results suggest that MFHR13 may benefit from the space between FH$_4$ and FH$_{19}$ to bind to C3b using both sites simultaneously, which could increase the decay of the convertases (C3bBb).

We demonstrated that FHR1, MFHR13, and the MFHR1 variants are able to bind not only to C5 and C5b6, but also to C6, C7, C8, and C9, which suggests that regulation is not only mediated through C5b6-complex binding, as currently assumed[15]. The interaction between MFHR13 and C5, C7, and C9 was confirmed in equilibrium conditions using MST. It was evaluated at increasing concentrations and the data showed dose-dependent binding of MFHR13 to all tested complement components. The concentration of stock solutions was 1 mg/ml. Based on the high intensity of binding, it was not possible to reach saturation of the binding curve for an accurate K$_D$ calculation with that concentration. Furthermore, taking into account that protein self-association via the N-terminal interaction domain (i.e., FHR1$_{1-2}$) also influences the strength of interaction by MST, the experimental values may not represent the true K$_D$. However, a K$_D$ value was extrapolated to ca. 4 µM for the interaction of MFHR13 with C3d. This estimation is in agreement with data

obtained in previous studies for the interaction between SCRs $FH_{19-20}$ and C3d, with values ranging 0.18 to 8 μM[56].

Our observations indicate an FHR1 activity resembling the function of vitronectin, which is a soluble human regulator of the MAC[57], and thus contribute to the mechanistic understanding of FHR1 action.

To distinguish the specific activity on C5 level from overall regulation, we evaluated the inhibition of the C5b-9 complex (MAC) formation on sheep erythrocytes in a serum-free approach using C5b6, C7, C8, and C9. We could demonstrate the activity of MFHR13 at this level of activation, which has to be attributed to the presence of the $FHR1_{1-2}$ as described previously[13,35].

C5 can be enzymatically activated by convertases or non-enzymatically by conformational changes when bound to highly C3b- or C4b-opsonized surfaces[5]. In contrast to enzymatic activation, conformational C5 activation cannot be prevented by eculizumab, explaining the residual hemolysis observed in PNH patients treated with adequate levels of this C5 inhibitor[5]. As opposed to eculizumab, MFHR13 could inhibit convertase-independent activation of C5 and MAC formation on C3b-opsonized sheep erythrocytes exposed to C5-C9. This mechanism of MAC formation, however, led to a weaker hemolysis than the one induced by C5b6-C9, even at higher concentrations of complement components in the assay, and its physiologic role should be further studied.

Several regulatory activities for FHR1 at the level of C5 at physiologic concentrations have been reported but the contribution of FHR1 in complement regulation is controversial[36,58], and the concentration of FHR1 in plasma is still under debate[59]. Moreover, FHR1 levels are considered to be elevated during infection, as suggested by increased expression of the FHR1 gene in zebrafish exposed to LPS[60,61]. Here, we used FHR1 and MFHR13 at a molar excess compared to the MAC components to inhibit C5b6-C9 or C5-C9 mediated hemolysis, however, we still used a concentration of FHR1 within its physiological range (0.1–2.7 μM)[59].

We consider that the SCRs $FHR1_{1-2}$ introduced important features to the fusion protein MFHR13. The regulatory activity at C5 level is fundamental and coordinates with FH activity at the level of C3. This is especially the case when the complement system has been activated by an infection, as observed in FHR1-homolog$^{-/-}$ mice, which exhibited higher C3a and C5a concentrations, severe inflammation and blood coagulation, leading to organ injury after LPS-induced complement activation[62].

We tested the overall regulatory activity in an ELISA-based assay, and in a cell-based assay by measuring the inhibition of hemolysis caused by the activation of the whole cascade. MFHR13 regulates the C5b-9 complex formation significantly more effectively than hFH when AP is activated by LPS in normal human serum. Although we could not demonstrate significant differences between MFHR1$^{I62}$ and MFHR13 in this assay, they outperformed eculizumab and FH activities by a factor of 2 and 37, respectively. Furthermore, MFHR13 protected sheep erythrocytes from hemolysis caused by complement attack, exhibiting 4-, 3-, and 26-fold the regulatory activity of hFH, MFHR1 versions, and eculizumab, respectively. Since the surfaces of sheep erythrocytes are rich in sialic acid, we assume that the higher binding efficiency of MFHR13 to polyanions might explain the higher local regulatory activity on cell surfaces.

SARS-CoV-2 spike proteins can bind to polyanionic glycosaminoglycans on cell surfaces and over-activate the AP, probably by displacing FH; however, exogenous FH was shown to protect cells from spike protein-induced complement attack[31]. Since MFHR13 binds to polyanions very efficiently and is a potent regulator of the AP, it might represent a promising alternative for COVID-19 therapy.

In summary, we designed MFHR13, a novel synthetic multi-target complement regulator, and produced it in moss bioreactors. MFHR13 is a glycoprotein that regulates the overall complement activity around 37 times as efficient as native human FH, and twice as much as eculizumab, without blocking the complement response completely. Taken together, MFHR13 is a potential therapeutic agent for diseases such as C3G and aHUS, and might also be of interest for different viral pathogenesis where an over-activation of the complement system is involved, such as dengue and COVID-19.

## Methods

**Comparative protein structure modeling approach.** To predict the 3D structure and C3b-binding capacity of a new multitarget complement regulator, MFHR13 ($FHR1_{1-2}:FH_{1-4}:FH_{13}:FH_{19-20}$), we used the template-based modeling implemented in Modeller 9.19[63].

High-resolution crystal and NMR structures for 17 of 20 domains of human FH are available in the Protein Data Bank (PDB). To build the model, we used the following structures as templates (PDB accessions): 3ZD2 ($FHR1_{1-2}$), 2WII ($FH_{1-4}$ + C3b), 2KMS ($FH_{12-13}$), 2G7I ($FH_{19-20}$), 3SW0 ($FH_{18-20}$), 2XQW ($FH_{19-20}$ + C3d), and a SAXS model for $FH_{11-14}$ (Accession in Small Angle Scattering Biological Data Bank (SASBDB): SASDAZ4). As the orientation between adjacent SCR domains is difficult to predict, some restrictions were added to orient the template structures appropriately. These restrictions were obtained by measuring distances Cα-Cα between adjacent SCRs in PDB structures such as 2WII, 3SW0 using the PyMOL Molecular Graphics System, Version 2.0, Schrödinger, LLC (PyMOL 2.0). Some additional restraints between $FH_4$ and $FH_{19}$ were taken into account, after superimposing the structures 2WII and 2XQW, which correspond to $FH_{1-4}$ with C3b and $FH_{19-20}$ with C3d, respectively.

The alignment of MFHR13 was done using also a structure for $FH_{11-14}$, derived from SAXS information (SASDAZ4). $FH_{14}$ was aligned with $FH_{19}$ because they share a sequence identity of 30%, and it would allow orienting $FH_{13}$. The module Automodel from modeler 9.19 was used to generate 100 models. The loop regions were refined using the loopmodel class, and 4 models with loop refined were built for every standard model (total: 400 models). Optimization of the objective function was done using 300 iterations with the conjugate gradient technique and molecular dynamics with simulated annealing to refine the model. The best models were chosen comparing the Discrete Optimized Protein Energy (DOPE) score. The quality of the models was evaluated using the Ramchandran Plot SAVeS Server (PROCHECK)[64] (https://services.mbi.ucla.edu/SAVES/), ProsaWEB (https://prosa.services.came.sbg.ac.at/prosa.php)[65]. 3D structures were visualized with PyMol 2.0.

**Generation of plasmid constructs.** First, the vector pAct5-MFHR1$^{I62}$, coding for MFHR1$^{I62}$, a MFHR1 variant with isoleucine instead of valine at position 193, which corresponds to position 62 of factor H, was created. This vector was obtained via site-directed mutagenesis using the Phusion Site-Directed Mutagenesis Kit (Thermo Fisher Scientific, Waltham, MA, USA) according to the manufacturer's instructions. The vector pAct5-MFHR1[41] was used as template with the primers MFHR1_I62_fwd and MFHR1_I62_rev (Supplementary Table 1), exchanging a single nucleotide (GTA to ATA, V62I). In this vector the expression of MFHR1$^{I62}$, fused to a C-terminal 8x His-tag, is driven by the 5′ region, including the 5′ intron, of the PpActin5 gene[66] and the cauliflower mosaic virus (CaMV) 35S terminator. For proper posttranslational modifications, the recombinant protein was targeted to the secretory pathway by the aspartic proteinase signal peptide from *P. patens*, PpAP1[67]. The hpt cassette[68] is present to select the transformed plants with hygromycin.

For the production of MFHR13 the expression construct pAct5-MFHR13, based on pAct5-MFHR1$^{I62}$, was generated using Gibson assembly[69]. For this, the sequence coding for SCR $FH_{13}$, together with its natural flanking linkers, was amplified from plasmid pFH[39] with the primers SCR13_overlapSCR4_fwd and SCR13_overlapSCR19_rev. This fragment was inserted in pAct5-MFHR1$^{I62}$, previously amplified with primers SCR4_overlapSCR13_rev and SCR19_overlapSCR13_fwd (Supplementary Table 1), designed to exclude the linker between $FH_4$ and $FH_{19}$ and overlapping the sequence of the linkers from $FH_{13}$. PCRs were performed with Phusion™ High-Fidelity DNA Polymerase (Thermo Fisher Scientific). Assembled vectors were verified by sequencing. The plasmids are available via the International Moss Stock Center IMSC (www.moss-stock-center.org/en/) under the accession numbers P1755 and P1762 for the expression constructs pAct5-MFHR1$^{I62}$ and pAct5-MFHR13, respectively.

**Plant material, culture, transformation procedure, and protein production.** Physcomitrella (*Physcomitrium patens*) was cultivated as previously described[68] on Knop ME medium[70]. Lines producing MFHR13 or MFHR1$^{I62}$ were obtained by stable transformation of the *Δxt/ft* moss line (IMSC no.: 40828), in which the genes coding for α1,3 fucosyltransferase and the β1,2 xylosyltransferase are knocked out[45].

Moss protoplast isolation, polyethylene glycol-mediated transfection (using 50 μg of linearized plasmid DNA), regeneration and selection were performed as previously described[68].

Plants surviving the selection were transferred to liquid medium and after 1 month of weekly propagation, moss lines were screened for the production of the proteins of interest. For this, 6-days-old moss suspension cultures were vacuum-filtrated and 30 mg fresh weight (FW) material were analyzed by ELISA. To analyze the time course of growth and production of the protein, lines were inoculated in triplicates at 200 mg/L DW and samples were taken every 3 days for 26 days.

The best MFHR1[I62] and MFHR13 producing moss lines, respectively, were scaled up to stirred tank bioreactors (5L), operated in batch for 8–12 days with the following conditions: pH 4.5, at 22 °C, aerated with 0.3 vvm air supplemented with 2% CO$_2$, stirred with 500 rpm, continuous light with an intensity of 160 μmol m$^{-2}$ s$^{-1}$ (day 0–2) and 350 μmol m$^{-2}$ s$^{-1}$ (day 2–8). The growth medium was supplemented with 1-naphthaleneacetic acid (10 μM NAA) at day 3 according to Top et al.[41].

**Purification of moss-produced recombinant proteins**. MFHR13 and MFHR1 variants (MFHR1[I62] and MFHR1[V62]), respectively, were extracted from vacuum-filtrated plant material. For this, 4 mL binding buffer (75 mM Na$_2$HPO$_4$, 0.5 M NaCl, 20 mM imidazole, 0.05% Tween-20, 10% glycerol, 1% protease inhibitor (P9599, Sigma-Aldrich), pH 7.0) were added per gram FW and the suspension was disrupted with an ULTRA-TURRAX® (10,000 rpm) and simultaneous sonication (ultrasonic bath) for 10 min on ice. After two consecutive centrifugation steps (4500 × g for 3 min and 20,000 × g for 10 min at 8 °C) the supernatant was filtered through 0.22 μm PES filters (Roth).

For chromatographic purification, the filtrate was loaded onto a 1 mL HisTrap FF column, using the ÄKTA system (Cytiva) at 1 mL/min. The column was washed with 30 column volumes (CV) of binding buffer and 3% elution buffer (100%: 100 mM Na$_2$HPO$_4$, 0.5 M NaCl, 500 mM imidazole, 10% glycerol, pH 7.4). The protein was eluted using a stepwise gradient (6% elution buffer for 10 CV, 17% 5 CV, 27% 3 CV, 100% 6 CV) and collected in 0.5 mL fractions. The first five fractions obtained with 100% elution buffer were pooled and diluted with Tris buffer pH 7.6 to reach 50 mM NaCl and loaded onto a 1 mL HiTrap Q HP column (Cytiva). MFHR13, MFHR1[I62], or MFHR1[V62] were eluted using a linear gradient (3–100%) and elution fraction containing the protein of interest (screened by ELISA, western blot, and Coomassie staining) were pooled and dialyzed against Dulbecco's phosphate-buffered saline (DPBS) in Slide-A-Lyzer® MINI Dialysis Devices, 20 K MWCO (Thermo Fisher Scientific). Proteins were concentrated by ultrafiltration using Vivaspin 2, 10 kDa MWCO (PES membrane; Sartorius).

**Protein quantification and immunoblotting**. MFHR13 and MFHR1 variants were quantified by ELISA using a modified protocol[39]: In order to quantify the fusion proteins using plasma-derived FH (hFH) standard as a reference, a polyclonal antibody against the whole FH was avoided, due to the differences in the protein structure and molecular weight between FH and the fusion proteins. Instead, a detection antibody against the FH$_{1-4}$, domains shared by all the proteins of interest, was used.

Microtiter plates (Nunc Maxisorp, Thermo Fisher Scientific) were coated overnight at 4 °C with GAU 018-03-02 (Thermo Fisher Scientific), a monoclonal antibody that recognizes FH$_{20}$ (1:2000 in coating buffer (1.59 g/l Na$_2$CO$_3$, 2.93 g/l NaHCO$_3$, pH 9.6)) and blocked with sample buffer (2% BSA in TBS (Tris Buffer Saline) supplemented with 0.05% Tween-20). Samples and hFH standard (hFH, CompTech; 12.9 pM–1.1 nM) were diluted in sample buffer and incubated for 90 min at 37 °C The proteins of interest were detected by a polyclonal anti-FH$_{1-4}$ (1:15,000 in washing buffer (1% BSA in TBS supplemented with 0.05% Tween-20)[71] and anti-rabbit coupled to horseradish peroxidase (HRP) (NA934; Cytiva, 1:5000 in washing buffer).

The ratio between MFHR13, MFHR1[I62], and MFHR1[V62] concentration was also compared with semi-quantitative western blots. SDS-PAGE on 7.5 or 10% gels, Coomassie staining and western blot analysis were performed as described before[41].

**Glycosylation analysis**. MFHR13, purified as described above, was used for MS analysis. In brief, duplicate samples of purified MFHR13 were reduced and alkylated as previously described[55], subjected to SDS-PAGE and subsequently stained with PageBlue® (Thermo Fisher Scientific). Bands corresponding to the expected size of MFHR13 were excised and destained. Digestion was performed overnight with 0.2 μg trypsin (Trypsin Gold, Promega) and 0.2 μg chymotrypsin (sequencing grade, Promega) simultaneously at 37 °C. Peptides were recovered and desalted using C18 StageTips (Thermo Fisher Scientific) and measured on a QExactive Plus Orbitrap (Thermo Fisher Scientific) as described in ref. [41]. Raw data were processed with Mascot Distiller V2.7.10 and a database search performed using Mascot server V2.7 (Matrix Science). Processed spectra from both duplicates were searched against a database containing all Physcomitrella protein models (V3.3[72]) as well as the sequence of MFHR13 and simultaneously against a database containing sequences of known contaminants (269 entries, available on request) using a precursor mass tolerance of 5 ppm and a fragment mass tolerance 0.02 Da. As variable modifications Gln-->pyroGlu (N-term. Q) −17.026549 Da, dehydration Glu->pyroGlu (N-term. E) −18.010565 Da, oxidation +15.994915 Da (M),

deamidation +0.984016 Da (N), GnGn +1298.475961 Da (N) were specified. Carbamidomethyl +57.021464 Da (C) was set as fixed modification. Search results were loaded in Scaffold5 software (Proteome Software, Inc.) and a threshold of 1% FDR at the protein level and 0.1% at the peptide level with a minimum of two identified peptides was specified.

Glycopeptides were identified from processed mgf files as described in ref. [55] using a custom Perl script. A list of searched glycopeptides is available in the Source data file. Quantitative values for identified glycopeptides were obtained from the allPeptides .txt file (available on request) from a default MaxQuant (V1.6.0.16) search on the raw data. All quantitative values were normalized against the sum of all precursor intensities from each raw file.

The mass spectrometry proteomics data have been deposited to the ProteomeXchange Consortium via the PRIDE[73] partner repository with the dataset identifier PXD025471 and https://doi.org/10.6019/PXD025471.

**Activity tests**

*Cofactor activity (CA)*. CA was measured in fluid phase. MFHR13, MFHR1[I62], MFHR1[V62], or hFH (15 or 75 nM) were incubated in DPBS with 444 nM C3b and 227 nM FI at 37 °C. Samples (20 μL) were collected at different reaction times up to 20 min. The reaction was stopped by the addition of 7.7 μL 4x Laemmli buffer (Bio-Rad) with 3 μL 50 mM DTT (NuPAGE, Thermo Fisher Scientific). Proteolytic cleavage of C3b was assessed by visualizing the α-chain cleavage fragments α'68 and α'43 by SDS-PAGE in 7.5% gels under reducing conditions by Coomassie staining. The bands corresponding to the intact C3b α'-chain were quantified by densitometry (GelAnalyzer 19.1; www.gelanalyzer.com) and normalized with the corresponding C3b-β-chain. The ratio α'-chain/β-chain at time 0 was set to 100% intact C3b α'-chain.

*Decay acceleration activity assay (DAA)*. The DAA in fluid phase was performed by an ELISA-based method as previously described[40]. Briefly, 250 ng C3b in PBS were immobilized on Maxisorp plates overnight at 4 °C. In order to generate the C3 convertases (C3bBb), 400 ng Factor B and 25 ng Factor D (CompTech) were incubated with immobilized C3b for 2 h at 37 °C. Increasing concentrations of moss-made regulators and hFH (Comptech) were added and incubated for 40 min at 37 °C to measure their ability to displace preformed C3 convertases. The Bb fragments that remain bound to C3b were detected by an anti-factor B polyclonal antibody (Merck, Darmstadt, Germany), followed by HRP-conjugated rabbit anti-goat (Dako, Hamburg, Germany). The absorbance of the preformed C3 convertase without regulators was set to 100% intact C3 convertases and the C3 proconvertase (C3bB) formation (FB without adding FD) was included as a negative control.

*Binding to complement proteins*. The ability of MFHR13 and MFHR1 variants to bind to the complement proteins was tested by ELISA. For this, 5 μg/mL of C3b, C5, C6, C7, C8, C9, or the complex C5b6 (CompTech, USA) in coating buffer were immobilized on Maxisorp plates at 4 °C overnight, blocked, and subsequently incubated with increasing concentrations of MFHR13, MFHR1[I62], or MFHR1[V62] (0.195–50 nM) in the case of testing C3b and C5 binding or a single concentration (25 nM or 50 nM) in the case of testing C6, C7, C8, C9, and C5b6 binding, diluted in sample buffer. Bound regulators were detected with anti-His-tag antibodies (MAB050, R&D Systems, 1:1000 in washing buffer) and HRP-conjugated anti-mouse IgG sheep (NA931, Cytiva, 1:5000 in washing buffer). In order to combine all independent experiments, the absorbance was normalized with the value corresponding to the highest concentration for every binding ELISA to obtain a relative binding.

Recombinant FHR1 with a C-terminal 6x His-tag (Abcam 152006) and hFH (Comptech, USA) were used as controls. However, due to the absence of His-tag in hFH, bound-hFH was detected with a polyclonal antibody[74] (1:1000 in washing buffer), and the HRP-conjugated anti-rabbit (1:5000 in washing buffer). It should also be considered, that the affinity of the anti-His-tag antibodies towards FHR1 compared to MFHR13, MFHR1[I62] and MFHR1[V62] might be different, due to the different lengths of the His-tags.

*Microscale thermophoresis (MST)*. The fusion protein MFHR13 (3 μM) was labeled with NT-647 RED-NHS (NanoTemper Technologies GmbH, Munich, Germany) as previously described[75]. NT-647-labeled MFHR13 (200 nM) was mixed with the ligands C5, C7, C9, and C3d, or BSA as control at dilutions ranging from 10 to 0.0005 μM. The samples were diluted in PBS supplemented with 0.05% Tween-20. Protein aggregation was minimized by centrifuging at 14,000 × g for 10 min. Thermophoresis was measured at 20% LED power and medium MST power in a Monolith NT.115 instrument (NanoTemper). All measurements were performed at RT using Premium Coated capillaries. Data were analyzed using MO.affinity Analysis Software (NanoTemper).

*Heparin binding*. Heparin-coated microplates (Bioworld, Dublin, Ohio, USA) were used to analyze binding of the regulators to this glycosaminoglycan (GAG) analog. Bound proteins were detected with anti-FH$_{1-4}$ (1:1000 in washing buffer) and HRP-conjugated anti-rabbit IgG from donkey (1:2000 in washing buffer, Cytiva).

*Overall AP regulatory activity*. The ELISA-based assay used to analyze the overall ability of the regulators to inhibit TCC formation after activation of the AP with lipopolysaccharides (LPS) was performed as described previously[35] with slight modifications. Briefly, increasing concentrations of MFHR13, MFHR1$^{I62}$, MFHR1$^{V62}$, eculizumab, or hFH (0.5–100 nM) were tested and formation of C5b-9 complex was detected using a C9 neoepitope-specific antibody (aE11, Santa Cruz Biotechnology, 1:2000 in DPBS/0.05% Tween-20), followed by HRP-conjugated anti-mouse IgG goat (NXA931, Cytiva; 1:5000 in DPBS/0.05% Tween-20). Samples with normal human serum (NHS) and without regulators were set to 100% AP activity. Heat-inactivated NHS (56 °C for 30 min) was used as a blank. A negative control to indicate spontaneous activation was included, which consisted of NHS without LPS and regulators.

The ability of the regulators to protect sheep erythrocytes from complement-mediated lysis was measured as follows: Increasing concentrations of MFHR13, MFHR1$^{I62}$, MFHR1$^{V62}$, eculizumab or hFH (0.3–100 nM) were incubated with $5 \times 10^7$ sheep erythrocytes followed by the addition of 30% FH-depleted serum (Comptech, USA) prepared in GVB/MgEGTA buffer (0.1% gelatin, 5 mM Veronal, 145 mM NaCl, 0.025% NaN$_3$, 5 mM MgCl$_2$, 5 mM EGTA, pH 7.3). The reaction was incubated for 30 min at 37 °C and stopped with GVB/EDTA (0.1% gelatin, 5 mM Veronal, 145 mM NaCl, 0.025% NaN$_3$, 10 mM EDTA). The amount of hemoglobin released was measured at 405 nm. Samples without regulators were set to 100% hemolysis, samples lacking NHS were included as negative controls and the values were subtracted from all samples.

*Regulation of MAC formation on sheep erythrocytes*. The inhibition of MAC formation on sheep erythrocytes was performed as previously described[13] with modifications. C5b6 (3.5 nM) was incubated with increasing amounts of the protein of interest (MFHR13, MFHR1$^{I62}$, MFHR1$^{V62}$, hFH, FHR1 (R&D Systems) or eculizumab, 1000 nM) for 10 min. Then a mixture of C7 (9 nM), C8 (0.667 nM), C9 (15 nM) and $5 \times 10^7$ sheep erythrocytes was added (prepared in GVB/MgEGTA buffer) in 50 µL total volume. Hemolysis was detected after 40 min at 37 °C by addition of GVB/EDTA buffer. The amount of hemoglobin released was measured at 405 nm. Samples without regulators were set to 100% MAC-induced lysis. BSA or purified extract from the parental line (*Δxt/ft*) were included as controls. Two negative controls without C5b6 or C9 were included, which were subtracted from all samples.

*Regulation of convertase-independent activation of C5 and MAC formation on sheep erythrocytes*. The ability of the regulators to inhibit convertase-independent activation of C5 and subsequent MAC formation was tested in a hemolytic assay. For this, C3b-opsonization on erythrocytes was carried out as previously described[76] with some modifications. To achieve maximal C3b deposition without significant MAC formation, FB was partially inactivated in FH-depleted serum at 50 °C for 5 min and 45 µL of this pretreated serum were added to 1 mL sheep erythrocytes ($10^9$/mL in GVB/EGTA-Mg$^{2+}$) and incubated at 37 °C for 35 min. Cells were washed 4 times with DPBS and resuspended in 1 mL GVB/EDTA buffer (Comp-Tech) to prevent formation of C3 convertases in case of residual FB.

C5 (75 nM) was preincubated for 15 min with MFHR13, FHR1, hFH, or eculizumab (700 nM). BSA, cytochrome c (Sigma 2037), or purified extract from the parental line (*Δxt/ft*) were included as controls. C3b-opsonized erythrocytes ($5 \times 10^7$ cells) together with C6 (110 nM) were added to the regulator mix and incubated for 10 min at 37 °C. Then, C7 (120 nM), C8 (70 nM), C9 (180 nM) were added to a total volume of 50 µL. After 45 min at 37 °C 100 µL GVB/EDTA were added and hemolysis was detected by measuring absorbance at 405 nm. Lysis without regulators was set to 100%. The negative control without C5 was subtracted from samples and controls.

**Statistics and reproducibility**. Analyses were done with the GraphPad Prism software version 8.0 for Windows (GraphPad software, San Diego, California, USA). For experiments involving a dose–response curve, logarithmic transformed data were fitted by a four-parameter logistic (4PL) nonlinear regression model to calculate the IC$_{50}$ and comparison of fits was carried out using the extra sum-of-squares F test with a cutoff at $P = 0.05$. Number of repeated experiments was indicated in the figure legends.

**Reporting summary**. Further information on research design is available in the Nature Research Reporting Summary linked to this article.

## Data availability

All data generated in this study are included in this paper and the supplementary information (uncropped gel and blot images are shown in Supplementary Fig. 11). In addition, a source data file is provided as Supplementary Data 1. Any remaining information can be obtained from the corresponding author upon reasonable request. The mass spectrometry proteomics data have been deposited to the ProteomeXchange Consortium via the PRIDE partner repository with the dataset identifier PXD025471 and https://doi.org/10.6019/PXD025471[77].

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

## Acknowledgements

This work was supported by the Deutscher Akademischer Austauschdienst DAAD (to N.R.M.), the Wissenschaftliche Gesellschaft Freiburg im Breisgau (to N.R.M.) and the Deutsche Forschungsgemeinschaft (DFG, German Research Foundation) under Germany's Excellence Strategy EXC-2189 (CIBSS; to R.R.). P.F.Z. acknowledges support from KIdnees Iowa City USA and from the Deutsche Forschungsgemeinschaft, Collaborative Research Center SFB1192, Immune mediated Glomerular Diseases, project B6. We thank Prof. Dr. Bettina Warscheid for the possibility to use the QExactive Plus instrument, Agnes Novakovic for excellent technical assistance, Andrea Hartmann for excellent assistance in performing and evaluating the MST experiments and Anne Katrin Prowse for proof-reading of the manuscript.

## Author contributions

N.R.M. performed most of the experiments, M.M. performed some of the binding tests, S.P. improved the purification process of the moss-made regulators, S.N.W.H. and L.L.B. performed the MS data analysis, P.F.Z. performed and interpreted MST. N.R.M., J.P., R.R., and E.L.D. designed the study and wrote the manuscript. P.F.Z. and K.H. discussed data and provided comments on the manuscript.

## Funding

## Competing interests

The authors are inventors of patents and patent applications related to the production of recombinant proteins in moss. R.R. is an inventor of the moss bioreactor and a founder of Greenovation Biotech, now eleva GmbH. He currently serves as advisory board member of this company.
