## [Transparent Peer Review File · Communications Biology]

Reviewer #1:

In this study, the authors engineered, produced, and characterized a chimeric protein regulator, MFHR13, which consisted of the C-5 regulatory domain of human FHR1 and the C3-regulatory domain of human FH. MFHR13 outperformed MFHR1, human FH and exulizumab in protecting sheep erythrocytes from complement-induced lysis.

The experiments were well designed and performed. I have a couple of comments on protein expression and quantification.

For a fair comparison among different proteins, the authors need to demonstrate how they quantified the purities and amounts of the proteins.

How did the authors choose moss as the host for MFHR13 production? E. coli and CHO are more commonly used in therapeutic protein production.

Reviewer #2:

Review of: A synthetic protein as efficient multitarget regulator against complement over-activation.

Summary:

The authors have generated a novel, recombinant complement regulator designed to inhibit the amplification of complement activation through the alternative pathway. It combines elements of the C5 regulator FHR1 and factor H to provide to inhibit the terminal pathway of complement activation. The authors use a plant-based expression system to produce the proteins, an IMAC purification method, and SDS-PAGE and mass-spectrometry to verify the purity and glycosylation status of the proteins produced. ELISA-based binding assays to examine the binding of MFHR1 polymorphisms, MFHR13, and Factor H to commercially obtained complement proteins. They then use fluid-phase C3 activation and SRBC hemolysis assays to measure the complement inhibiting capability of MFHR13 and discuss its potential use as a novel therapeutic agent to modulate complement activation in multiple disease settings.

Major Comments:

A major concern in this paper is the design and interpretation of the ELISA-based binding experiments. One interpretation of the results is that MFHR13 (and its parental protein FHR1) show a promiscuous binding specificity toward C5, C6, C7, C8, and C9. An alternative interpretation is that the observed binding of MFHR13 to one or more of these proteins is an artifact of protein aggregation and a method that obscures the true relative affinities of MFHR13 for these proteins. To rule out the latter interpretation, technical changes to the experimental design are needed that would allow more accurate quantification of the proposed binding interactions to ascertain which interactions are physiologically relevant. This will be needed to avoid incorrect conclusions and provide data and mechanism of action.

Protein preparation: The analyte proteins in these assays are purified by IMAC and ion-exchange chromatography, but no efforts are made to remove misfolded proteins or protein aggregates prior to analysis. Protein aggregates frequently lead to enhancement of protein avidity, not only for native ligands, but also for weak and non-specific interactions in intermolecular binding assays. Authors should perform size exclusion chromatography on all analytes immediately prior to binding assays to remove protein aggregates and limit non-specific interactions.

Binding analysis: ELISA is an inappropriate method for quantification of relative binding affinities of most proteins, and the normalized ELISAs performed in this paper are especially hazardous. This is because, in order to estimate affinity from an equilibrium binding analysis, binding must be measured

at equilibrium with both reactants present. In the ELISA assays shown, binding is measured by a detection antibody well after analytes have been removed from the immobilized ligand. This situation favors measurement of residual protein aggregate deposits late in the dissociation phase rather than the equilibrium or kinetics of a first-order reversible binding interaction as it occurs. In this case, the signal observed may be proportional to the level of analyte aggregation rather than physiological binding affinity. This will result in misleading data and incorrect interpretations. The promiscuous binding of MFHR13 to all members of the terminal complement pathway along with strong associations with polyanions is consistent with aggregation, and therefore confounds the production of mechanistic conclusions. The ELISA method also precludes determination of equilibrium and/or kinetic binding constants that are needed for valid quantitative comparisons between polymorphic proteins or between MFHR13 and other regulators.

For all quantitative binding assays where an ELISA has been used, authors should use a method such as surface plasmon resonance, isothermal calorimetry, or biolayer interferometry to for quantitation, and use the produced binding constants to draw comparisons and mechanistic conclusions. A JBC article by Katschke et al (doi: 10.1074/jbc.M112.345082) provides an example of how this might be performed as well as valuable kinetic data that show the importance of measuring these dynamic complexes at equilibrium rather than late in the dissociation phase. If the authors do not have access to this instrumentation, they should look at an article from Thomas Pollard in MBoC (doi: 10.1091/mbc.E10-08-0683) for guidance on how ELISA combined with other methods can be used for quantifying interactions measured at equilibrium.

Minor comments:

In the introduction, the authors write that, "the contribution of FH13 is still discussed," with reference 9 indicated. Expanding this statement to include the potential contributions of FH13 would be helpful.

Figure 2 and other binding data: When performing these binding assays under equilibrium binding conditions, the most appropriate regression model is usually a one-site binding model, which will deliver a K_d value.

Figure 2: The authors indicate that MFHR1-I62 displays stronger binding and cofactor activity than the V62 variant, but this is not evident from the representative figure shown. In addition, the statistics derived from just two independent experiments do not make a strong case for I62 superiority. This experiment should be replicated to better ascertain whether a biologically significant functional difference exists between the V and I polymorphisms.

Figure 7 (and others). The use of relative binding on the panels obscures the absolute magnitude observed and precludes stoichiometric analysis. Using a method such as SPR would greatly enhance the mechanistic data available.

Figure 7E: The authors acknowledge that the enhancement of hemolysis by eculizumab was unexpected. If this was a recurring phenomenon, the authors should offer a potential mechanistic explanation in the discussion.

Point-by-point response to the reviewers:

- Reviewer #1:

In this study, the authors engineered, produced, and characterized a chimeric protein regulator, MFHR13, which consisted of the C-5 regulatory domain of human FHR1 and the C3-regulatory domain of human FH. MFHR13 outperformed MFHR1, human FH and exulizumab in protecting sheep erythrocytes from complement-induced lysis.

The experiments were well designed and performed. I have a couple of comments on protein expression and quantification.

For a fair comparison among different proteins, the authors need to demonstrate how they quantified the purities and amounts of the proteins.

Reply:

We thank the reviewer for the positive evaluation of our work.

One of the critical points in our study was the correct determination of protein concentrations. The quantification was carried out using ELISA. For the standard curve (molarity), we used plasma-derived FH (hFH) (Comptech, USA) and a detection antibody against the SCRs1-4, domains shared by all the proteins of interest.

We confirmed the results using a semi-quantitative Western blot (anti-His or anti-SCR1-4), using as standard curve the values determined by ELISA to make sure that the ratio or proportion between the purified proteins were correct. (In Figure 4B we show the western blot with anti-His antibodies, which is similar to the Western blot with anti-SCR1-4 antibodies.)

The purity of the sample was evaluated via PageBlue™ Coomassie staining, which stains proteins in a sensitive and quantitative way. Although some faint contaminants could be detected, the proteins of interest are by far the most abundant in the sample (Figure 4A).

- How did the authors choose moss as the host for MFHR13 production? E. coli and CHO are more commonly used in therapeutic protein production.

Reply:

The moss *Physcomitrium patens* (Physcomitrella) has a high efficiency of gene targeting via homologous recombination and is recognized a model organism for cell biology and developmental studies. It was the first non-vascular plant to have the complete genome sequenced. Our research group has pioneered applied research

using mosses, especially the production of biopharmaceuticals to treat human diseases (Decker and Reski 2020, doi.org/10.1016/j.copbio.2019.09.021).

The first recombinant production of human Factor H was achieved in this biological system in significant amounts to complete a preclinical study (Buttner-Mainik et al 2011, doi.org/10.1111/j.1467-7652.2010.00552.x, Michelfelder et al 2017, doi.org/10.1681/ASN.2015070745). No medication is currently available to control complement activation on the level of the central compound C3. Therefore, we targeted our efforts to produce complement therapeutics in moss bioreactors, as a niche to exploit this production platform.

Low-cost culture medium, high batch-to-batch stability and glyco-engineering for improved pharmacokinetics and pharmacodynamics of the biopharmaceutical are some of the advantages of this system we can highlight. Furthermore, α -galactosidase produced in moss (Repleva aGal; eleva GmbH) to treat Fabry disease (Hennermann et al 2019, doi.org/10.1002/jimd.12052), showed superior characteristics compared to the mammalian cell-based protein and successfully completed clinical trial phase I.

- Reviewer #2:

Review of: A synthetic protein as efficient multitarget regulator against complement over-activation.

Summary:

The authors have generated a novel, recombinant complement regulator designed to inhibit the amplification of complement activation through the alternative pathway. It combines elements of the C5 regulator FHR1 and factor H to provide to inhibit the terminal pathway of complement activation. The authors use a plant-based expression system to produce the proteins, an IMAC purification method, and SDS-PAGE and mass-spectrometry to verify the purity and glycosylation status of the proteins produced. ELISA-based binding assays to examine the binding of MFHR1 polymorphisms, MFHR13, and Factor H to commercially obtained complement proteins. They then use fluid-phase C3 activation and SRBC hemolysis assays to measure the complement inhibiting capability of MFHR13 and discuss its potential use as a novel therapeutic agent to modulate complement activation in multiple disease settings.

Major Comments:

A major concern in this paper is the design and interpretation of the ELISA-based binding experiments. One interpretation of the results is that MFHR13 (and its parental protein FHR1) show a promiscuous binding specificity toward C5, C6, C7, C8, and C9. An alternative interpretation is that the

observed binding of MFHR13 to one or more of these proteins is an artifact of protein aggregation and a method that obscures the true relative affinities of MFHR13 for these proteins. To rule out the latter interpretation, technical changes to the experimental design are needed that would allow more accurate quantification of the proposed binding interactions to ascertain which interactions are physiologically relevant. This will be needed to avoid incorrect conclusions and provide data and mechanism of action.

Binding analysis: ELISA is an inappropriate method for quantification of relative binding affinities of most proteins, and the normalized ELISAs performed in this paper are especially hazardous. This is because, in order to estimate affinity from an equilibrium binding analysis, binding must be measured at equilibrium with both reactants present. In the ELISA assays shown, binding is measured by a detection antibody well after analytes have been removed from the immobilized ligand. This situation favors measurement of residual protein aggregate deposits late in the dissociation phase rather than the equilibrium or kinetics of a first-order reversible binding interaction as it occurs. In this case, the signal observed may be proportional to the level of analyte aggregation rather than physiological binding affinity. This will result in misleading data and incorrect interpretations. The promiscuous binding of MFHR13 to all members of the terminal complement pathway along with strong associations with polyanions is consistent with aggregation, and therefore confounds the production of mechanistic conclusions. The ELISA method also precludes determination of equilibrium and/or kinetic binding constants that are needed for valid quantitative comparisons between polymorphic proteins or between MFHR13 and other regulators.

For all quantitative binding assays where an ELISA has been used, authors should use a method such as surface plasmon resonance, isothermal calorimetry, or biolayer interferometry to for quantitation, and use the produced binding constants to draw comparisons and mechanistic conclusions. A JBC article by Katschke et al (doi: 10.1074/jbc.M112.345082) provides an example of how this might be performed as well as valuable kinetic data that show the importance of measuring these dynamic complexes at equilibrium rather than late in the dissociation phase. If the authors do not have access to this instrumentation, they should look at an article from Thomas Pollard in MBoC (doi: 10.1091/mbc.E10-08-0683) for guidance on how ELISA combined with other methods can be used for quantifying interactions measured at equilibrium.

Reply:

We thank the referee for his thorough review and advice how to improve our manuscript.

The aim of our work was to design and produce a new potent multitarget complement regulator with high overall regulatory activity *in vitro* and analyze if our new synthetic protein MFHR13 displays the expected activities derived from the physiological proteins without any restraint. Therefore, it was essential to compare the complement-regulatory activity of MFHR13 to that of its native progenitors, FH and FHR1. We agree with the reviewer that the results obtained via ELISA for the interaction of MFHR13 with the compounds forming the Terminal Complement Complex should be confirmed as these interactions were shown for the first time for any complement regulator. Therefore, we repeated the interaction study in equilibrium as the reviewer suggested, using Microscale Thermophoresis (MST).

Being a relatively new technique (available since 2010) MST is becoming the first option to characterize biomolecular interactions due to its advantages, such as the possibility to measure interactions in complex solutions similar to native conditions, low sample consumption, and the immobilization-free setup.

We analyzed the binding between MFHR13 and four of the ligands: three components of the MAC (C5, C7, C9), and C3d as a positive control. We could verify that the interaction between MFHR13 and these components of the MAC takes place in equilibrium with both reactants present, and therefore validated the ELISA results shown. We added the results as new Figure 7D in the revised version of our ms. Although we used the highest possible concentration for the commercial proteins (C5, C7, C9, and C3d), it was not possible to reach saturation, which affects the estimation of an accurate equilibrium dissociation constant (K_D). However, the estimated K_D for these interactions are in the micromolar range.

Furthermore, we chose the C3d domain as positive control due to the monovalency of its interaction with complement regulators; thus we assume a 1:1 interaction model for all tested analytes. The estimated K_D for the interaction between C3d and MFHR13 is around 4 μ M, and is congruent with values reported in the literature for SCR19-20 of FH and C3d, which range between 0.18 and 8 μ M (Perkins et al 2012, doi.org/10.1016/j.imbio.2011.10.003).

With the MST affinity studies presented in the revised version of our ms we have proven that there is true interaction of MFHR13 with the complement compounds.

We abstained from additional confirmation of ELISA-generated results for the following reasons:

- As mentioned above it was our aim to present the suitability of MFHR13 to fight complement-associated diseases, thus to analyze its interaction with putative interaction partners, but not to quantify binding affinities. This misunderstanding was partially caused by ourselves by inaccurate wording which we corrected now in the revised version of the ms.

- ELISA is a widely accepted methodology to evaluate biomolecular interactions of complement proteins and provides a platform of comparative results. (The method is included in the review doi: 10.1002/cmdc.201500495, *ChemMedChem* 2016). We specify some examples from the literature below to support our arguments.
- We think that ELISA, SPR, BLI, and MST are complementary and valid methods. In contrast to ELISA, SPR, BLI and MST are more sensitive and allow the calculation of association and dissociation rates, or equilibrium constants, valuable information to understand the mechanism of action. We agree with the reviewer that it is useful to calculate equilibrium dissociation constants when it is practical, but due to the arguments presented in the following, in our case it is not.

The interaction of our protein MFHR13 or FH with C3b and heparin is multivalent (they bind to C3b through SCR 1-4 and SCR 19-20, and to heparin through SCR13, SCR20 and additionally SCR7 in FH). So far, an efficient label-free method to determine multivalency protein binding is not available. Surface plasmon resonance (SPR), which is considered a standard technique to analyze protein interactions, cannot distinguish multivalent from monovalent binding and the fidelity of the measurement is compromised. Furthermore, we confirmed in a previous study (Michelfelder et al 2018, doi.org/10.1681/ASN.2017070738) that MFHR1 occurs predominantly as dimers or multimers, while it is estimated that 5-14% of FH is oligomeric. Therefore, there are discrepancies in the literature in K_D values measured for FH interactions with its ligands, since these aspects (oligomerization and multivalency) are not taken into account. In our case, MFHR1 or MFHR13-ligand (C3b and heparin) interactions represent an even more complex system in comparison to FH, which makes more challenging an accurate quantitative kinetic analysis. MST is also affected by self-association, therefore, the apparent K_D calculated by SPR, Biolayer Interferometry (BLI), or MST can be many orders of magnitude different from the true K_D . Although, some approaches to overcome these limitations have been proposed recently, this is out of the scope of our study (Yang et al 2020, doi.org/10.1021/acsami.9b17328).

Yang et al. (2018) tried to determine the binding kinetic profiles and K_D for the interaction of C3b and a complement regulator with the dimerization interface derived from FHR1 using SPR. However, they failed to calculate it due to the complexity of the system (Yang et al 2018, doi.org/10.1681/ASN.2017091006).

Examples for ELISA-based results about complement interaction studies are:

Comparative binding of FH variants V62 and I62 to C3b, confirmed by SPR (Tortajada et al 2009, doi.org/10.1093/hmg/ddp289, figure 1 and 2).

Comparative binding of FH and a truncated FH version to C3b and heparin (Nichols et al 2015, doi.org/10.1038/ki.2015.233_figure 1b,c)

Binding of FHR4 isoforms to C3b (Hebecker and Józsi 2012, doi.org/10.1074/jbc.M112.364471, figure 2a)

Comparative binding of increasing concentrations of pentameric and monomeric C-reactive protein to immobilized FHR proteins (Csincsi et al 2017, doi.org/10.4049/jimmunol.1600483, figure 3).

Binding of properdin to immobilized granulocyte myeloperoxidase (Harboe et al 2017, doi.org/10.1073/pnas.1612385114, figure 1).

Binding of FHR1 to Malondialdehyde-modified low-density lipoproteins, confirmed by SPR (Irmscher et al 2019, doi.org/10.1038/s41467-019-10766-0, figure 6 a-c)

Comparative binding of engineered anti-SARS-CoV-2 antibodies to viral spike protein (Leach et al 2021, doi.org/10.1038/s41598-021-89887-w, figure 3).

Our own studies about comparative binding of MFHR1 and FHR1 to C5 (Michelfelder et al 2018, doi.org/10.1681/ASN.2017070738, figure 4a and supplement) and about comparative binding of moss-produced FH vs. plasma-derived FH to C3b and heparin (Michelfelder et al 2017, doi.org/10.1681/ASN.2015070745, figure 2A, B).

We consider our methodology adequate to conclude that our new synthetic protein MFHR13 retains the ability to bind to the most important ligands C3b and heparin in comparison to MFHR1.

We included normalized values in the ELISA to present all data together, since at least three independent experiments were performed. For the sake of clarity, we added a representative experiment with the absorbance values obtained as supplementary information (Supplementary Fig. 7, 8, and 9). The IC₅₀ calculated after non-linear regression is the adequate way to determine statistically significant differences between the binding curves. Furthermore, we deleted comparison in figure 7c, in which we used ELISA with a single concentration.

- Protein preparation: The analyte proteins in these assays are purified by IMAC and ion-exchange chromatography, but no efforts are made to remove misfolded proteins or protein aggregates prior to analysis. Protein aggregates frequently lead to enhancement of protein avidity, not only for native ligands, but also for weak and non-specific interactions in intermolecular binding assays. Authors should perform size exclusion chromatography on all analytes immediately prior to binding assays to remove protein aggregates and limit non-specific interactions.

Reply:

To compare MFHR13 with other MFHR1 variants, we produced, purified and quantified these proteins always following strictly the same procedure. We included

two chromatography steps (IMAC (Ni-NTA) and anion exchange) to get rid of host protein contaminants and aggregates. The second chromatography step was not part of the procedure in our previous study, in which MFHR1 was just purified by IMAC (Michelfelder et al. 2018, doi.org/10.1681/ASN.2017070738). In that study, size exclusion chromatography was performed after purification to analyze the distribution of MFHR1 molecular species. We observed that MFHR1 migrates mainly in a multimeric state, similar to FHR1 in human serum. Multimeric MFHR1 fractions exhibit higher inhibitory activity than monomeric fractions, in contrast to FH, which migrates mainly in a monomeric form. The ability of these synthetic proteins to form dimers or multimers is one of the most important advantages as a potential biopharmaceutical to increase the potency of the drug, by enhancing the overall regulatory activity.

Size exclusion chromatography (SEC) is not easily conducted at industrial scale for biopharmaceuticals production; however, anion exchange chromatography can be used to separate aggregates, because they normally have a higher surface charge. For the elution in our anion exchange chromatography, we used a linear gradient from 78 mM to 1 M NaCl in 10 column volumes (10 mL), and collected the elution in 1 mL fractions. MFHR13 started eluting above 420 mM NaCl (end of fraction 3) and was detected until fraction 10 in decreasing concentrations, which might indicate that the protein can form aggregates, which elute later. As fractions 3 and 4 showed the best ratio of product concentration and purity we decided to proceed with exclusively these fractions. According to our methodology we expect to have reduced degree of aggregation. Furthermore, we recovered just around 6% of the protein at the end of the purification process, and inclusion of a third chromatography step such as SEC would further reduce the yield.

Moreover, using SEC immediately before binding experiments is not possible and practical for us because, apart from a drop in yield, the sample will be diluted. The highest concentrations we achieved are between 200 and 500 µg/ml for MFHR13 and MFHR1 variants. Therefore, it would be necessary to concentrate the sample once more and exchange the buffer, where the putative formation of aggregates cannot be excluded.

For the assays we used samples from different purification batches and we always got similar results. If the binding would be significantly affected by aggregate formation, we would expect a high variation in every binding ELISA.

Minor comments:

In the introduction, the authors write that, “the contribution of FH13 is still discussed,” with reference 9 indicated. Expanding this statement to include the potential contributions of FH13 would be helpful.

Reply: We added the following information to the revised version of the ms: “The contribution of FH₁₃ has been a matter of discussion (Blackmore et al. 1998; Pangburn et al. 1991; Schmidt et al. 2010). Recently, a heparin binding site within fragment FH₁₁₋₁₃ was reported with an affinity constant around 17 μ M, 4 and 14-fold weaker than that for FH₂₀ and FH₇, respectively, measured by Surface Plasmon Resonance (SPR) (Haque et al 2020). FH₁₃ has special structural characteristics which indicates a functional role, for instance, it is the smallest domain in FH with a positive net charge (+5), similar to FH₇, and contains the longest linkers in FH (Schmidt et al. 2010).”

Figure 2 and other binding data: When performing these binding assays under equilibrium binding conditions, the most appropriate regression model is usually a one-site binding model, which will deliver a K_d value.

Reply: As mentioned above, the interaction of MFHR1 and MFHR13 to C3b and heparin is not monovalent, therefore a one-site binding model would not be accurate. Furthermore, these proteins mainly form dimers, which impedes the estimation of a realistic K_D.

Figure 2: The authors indicate that MFHR1-I62 displays stronger binding and cofactor activity than the V62 variant, but this is not evident from the representative figure shown. In addition, the statistics derived from just two independent experiments do not make a strong case for I62 superiority. This experiment should be replicated to better ascertain whether a biologically significant functional difference exists between the V and I polymorphisms.

Reply: It was reported previously that the plasma-purified FH I62 variant in comparison to the V62 variant showed increased binding to C3b and enhanced cofactor activity for FI-mediated C3b cleavage (Tortajada et al 2009, doi.org/10.1093/hmg/ddp289). Therefore, our aim was to confirm the superiority of I62 with the two existing MFHR1 variants, V62 and I62, before we introduced I62 in the new complement regulator MFHR13. Although we could confirm increased cofactor activity for MFHR^{I62}, a higher difference in the regulatory activity of both FH variants purified from plasma had been observed before. The substitution of valine for isoleucine occurs as a result of a single nucleotide polymorphism and both amino acids are non-polar, therefore it is unlikely to have a deep effect on the overall regulatory activity of FH. However, the additional methylene group from the isoleucine substitution triggers a small rearrangement in the SCR1 of FH and it is slightly more thermally stable than the variant V62. We considered that two independent experiments with three different reaction times each were sufficient to confirm previous results for the cofactor assay.

Figure 7 (and others). The use of relative binding on the panels obscures the absolute magnitude observed and precludes stoichiometric analysis. Using a method such as SPR would greatly enhance the mechanistic data available.

Reply: Please see our answer above. We confirmed the results shown in figure 7C using a new technique, Microscale thermophoresis (MST). We added Supplementary Fig. 7 in the revised version of our ms to show a representative experiment with the obtained absorbances, i.e. presenting the absolute magnitude.

Figure 7E: The authors acknowledge that the enhancement of hemolysis by eculizumab was unexpected. If this was a recurring phenomenon, the authors should offer a potential mechanistic explanation in the discussion.

Reply: Binding of eculizumab can block the proteolytic activation of C5. However, this binding cannot avoid C5 activation via its conformational change on cell surfaces with a high C3b density. Our study addressed the activity of MFHR13 in decrease of hemolysis and used eculizumab as a negative control. A mechanistic explanation of the increase of hemolysis in the sample with eculizumab would require a deeper study on the antibody, what was not our purpose. One could speculate: The increase in hemolysis observed in the samples with eculizumab is likely to be independent of C5 due to higher fragility of C3b-opsonized erythrocytes. It is feasible that the formulation of Soliris leads to higher hemolysis compared to our control without regulators. Polysorbate 80 (0.22 mg/ml) is present in the formulation of Soliris. Although this is a very common non-anionic surfactant included in the formulation of many protein biopharmaceuticals, this compound has been shown to trigger adverse effects such as hemolysis (Sun et al 2017, DOI: 10.1039/C6RA27242H). Despite its low concentration in Soliris it might trigger hemolysis due to osmotic pressure in highly opsonized erythrocytes which have a high osmotic fragility.

REVIEWERS' COMMENTS:

Reviewer #1 (Remarks to the Author):

I recommend the acceptance of this revised manuscript.

Reviewer #3 (Remarks to the Author):

The authors have addressed, through experiments or rebuttal, the technical concerns brought up in the prior review.

I would caution the authors that the low production yield, potential for aggregation, and unexplained broad specificity are red flags for the suitability of this protein in a therapeutic setting. Its potential impact would have been greatly increased through a functional evaluation in vivo.

Response to Referees:

We thank both Referees for their comments. We will work on lines with higher product yields and in vivo studies.

Reviewer #1 (Remarks to the Author):

I recommend the acceptance of this revised manuscript.

Reviewer #3 (Remarks to the Author):

The authors have addressed, through experiments or rebuttal, the technical concerns brought up in the prior review.

I would caution the authors that the low production yield, potential for aggregation, and unexplained broad specificity are red flags for the suitability of this protein in a therapeutic setting. Its potential impact would have been greatly increased through a functional evaluation in vivo.